

COMPUTO

ISSN 2824-7795

# Comparative analysis of stopping criteria for multi-objective evolutionary algorithms: from benchmark problems to industrial application

Manon Perrignon [1]    L'Institut Agro, INRAE, STLO (Science et Technologie du Lait et de l'œuf), Rennes, France

Magalie Houée-Bigot    L'Institut Agro, Rennes, France

Romain Jeantet    L'Institut Agro, INRAE, STLO (Science et Technologie du Lait et de l'œuf), Rennes, France

Thomas Croguennec    L'Institut Agro, INRAE, STLO (Science et Technologie du Lait et de l'œuf), Rennes, France

Mathieu Emily [2]    L'Institut Agro, Université de Rennes, CNRS, IRMAR (Institut de Recherche Mathématique de Rennes)-UMR 6625, Rennes, France

Date published: 2026-06-23    Last modified: 2026-06-23

## Abstract

Multi-objective optimization problems are commonly solved using evolutionary algorithms, which typically terminate after a user-defined number of generations without guaranteeing solution optimality. Stopping criteria can address this limitation by determining when the algorithm has converged to high-quality solutions. This study compares four existing stopping criteria and proposes a novel criterion designed to balance computational cost and solution quality. The criteria are first evaluated on classical multi-objective benchmark problems, then applied to test cases where optimal solutions are known and can be compared with those obtained after implementing each stopping criterion. Finally, the criteria are applied to a food process optimization case study. This analysis identifies the most effective criterion for achieving this trade-off in practical applications.

*Keywords:* Multi-objective optimization, Stopping criteria, Evolutionary algorithms, Evaluation metrics, Food process optimization

# Contents

[1]Corresponding author: manon.perrignon@institut-agro.fr
[2]Corresponding author: mathieu.emily@institut-agro.fr

# 1 Introduction

Industrial performance is a key issue for companies, particularly in the food sector, where turnover is usually very low and the sector is highly challenging. It reflects a production system's ability to meet its quality, cost, sustainability and environmental impact objectives (Madoumier et al. 2019). Industrial performance is a multifaceted concept assessed through a variety of interdependent and sometimes contradictory indicators. Therefore, optimizing performance means finding a balance between multiple objectives, adding complexity to the decision-making process (Drofenik et al. 2023).

To address this challenge, multi-objective optimization methods provide a rigorous framework for identifying trade-offs between different performance objectives (Ehrgott 2005; Wari and Zhu 2016). The aim of this method is to find a set of solutions that meet the defined objectives, known as the Pareto front. Evolutionary algorithms are commonly used to generate the Pareto front. This family of algorithms is inspired by evolutionary theories (Konak et al. 2006), with notable examples including NSGA-II (Non-dominated Sorting Genetic Algorithm II) (Deb et al. 2002) and SPEA-2 (Strength Pareto Evolutionary Algorithm 2) (Zitzler et al. 2001).

For most algorithms, the stopping criterion corresponds to the maximum number of generations to be performed, which is set *a priori* by the user. However, this predefined number of generations does not guarantee the quality of the solutions if it is too low or may lead to unnecessary consumption of computational resources if it is too high. In the latter case, the algorithm continues processing even when improvements are minimal or non-existent (Roudenko and Schoenauer 2004). An effective stopping criterion would be to find the minimum number of generations beyond which the algorithm cannot improve the quality of the solutions. The quality of the Pareto front is usually measured using two main categories of metrics: diversity metrics and convergence metrics (Li and Yao 2020). Diversity metrics assess how well the solutions are distributed along the Pareto front, ensuring adequate coverage of the objective space. Their goal is to avoid clustering in specific regions and to promote a large and uniform exploration of the objective space. Examples of metrics include spacing measures and entropy-based metrics. Convergence metrics, on the other hand, either evaluate how close the current solutions are to the theoretical Pareto front, when this is known in advance; or evaluate the extension of the Pareto front with respect to a reference point in objective space. Common metrics in this category include hypervolume and generational distance (Liu et al. 2018).

Hypervolume measures the space covered by solutions in the objective space and generational distance measures the distance of the front from a reference front (Audet et al. 2021). To guarantee the quality of the solutions when the algorithm stops, a stopping criterion based on the evaluation metrics of the solutions is required. In this regard, several stopping criteria based on the quality of the solutions have been proposed in the literature.

A typical stopping criterion contains two main components: an evaluation metric and a termination criterion. The evaluation metric measures the evolution of the solutions, while the termination criterion determines whether the algorithm should stop depending on the evaluation metric (Liu et al. 2018). For comparative purposes, four criteria frequently used in the literature were selected. These criteria are defined based on various evaluation metrics and termination criteria. The MGBM criterion belongs to the class of convergence based metric with the Mutual Domination Rate (MDR) coupled with a Kalman filter (Martí et al. 2016). The OCD HV (Online Convergence Detection based on Hypervolume) criterion relies on a Chi-squared and the hypervolume convergence metric to evaluate the algorithm's progress (Wagner et al. 2009). The advantage of this criterion is its reduced computational complexity, although according to the literature, it tends to stop too early (Wagner et al. 2009). The LSSC (Least Square Stopping Criterion) is based on the analysis of residuals from a linear regression applied to the evolution of an evaluation metric over a window of generations. In this paper, the chosen metric is hypervolume. This criterion aims to be easy to implement (Guerrero et al. 2010). Finally, the entropy-based criterion (Population-based entropy) calculates a dissimilarity measure based on entropy between solution populations across generations (Saxena et al. 2016).

Stopping criteria have already been compared on standard multi-objective problems commonly found in the literature (Abu Doush et al. 2023; Martí et al. 2016). These comparative studies highlight that there is no single optimal stopping criterion suitable for all algorithms and problem types. OCD HV criterion has been shown to stop earlier than others, while MGBM can suffer from significant computational cost. The effectiveness of a given criterion can be influenced by the specific characteristics of the problem being addressed (Abu Doush et al. 2023). Moreover, some criteria may become computationally expensive as the number of objectives increases. This is the case with the hypervolume metric (Liefooghe and Derbel 2016). Additionally, the literature demonstrates that the quality of a Pareto front cannot be adequately assessed by a single evaluation metric (Tsarmpopoulos et al. 2019; Halim et al. 2021). Therefore, it is crucial to develop more robust and adaptive stopping criteria, for example by combining multiple evaluation metrics of the Pareto front.

In this paper, we introduce a new stopping criterion, called MPF for Maximum Performance Front. MPF aims to leverage two evaluation metrics and will be evaluated against existing criteria. Although benchmark problems are designed to match real-world applications, their formulations remain far from industrial challenges, where objective functions are often unknown and highly complex. In an industrial context, a stopping criterion is essential to be able to use the multi-objective optimization method more efficiently. Presently, stopping criteria have never been compared on industrial applications and deserve to be evaluated in such contexts (Abu Doush et al. 2023).

The paper is organized as follows: Section 2. presents the existing criteria and introduces MPF criterion. Section 3. describes the methodology used to compare stopping criteria and the different multi-objective problems used. Section 4. presents the results obtained for each approach, followed by a final section where the results are compared and discussed.

## 2 Stopping criteria for multi-objective optimization

### 2.1 Definition of evaluation metrics

The quality of solutions produced by a multiobjective optimization algorithm can be quantified using evaluation metrics. These metrics fall into two broad categories: convergence metrics, which measure how closely the Pareto front approximation approaches the optimal front and diversity metrics, which assess the distribution and spread of solutions along the front.

Convergence metrics evaluate the extent to which the approximated solution set approaches the reference Pareto front or progresses from one iteration to the next. The hypervolume (Guerreiro et al. 2022) is one of the most widely used metrics in multiobjective optimization. It measures the volume of the objective space dominated by the Pareto front approximation $Y_N$ and bounded by a reference point $\mathbf{r}$:

$$HV(Y_N; \mathbf{r}) = \lambda_k \left( \bigcup_{\mathbf{y} \in Y_N} [\mathbf{y}, \mathbf{r}] \right) \tag{1}$$

**where**:

- $Y_N$: Pareto front approximation.
- $k$ : number of objectives
- $\mathbf{r} \in \mathbb{R}^k$: reference point, such that for all $\mathbf{y} \in Y_N$, $\mathbf{y} \leq \mathbf{r}$
- $\lambda_k$: $k$-dimensional Lebesgue measure

A higher hypervolume value indicates a better approximation of the Pareto front. This metric is particularly valued because it is sensitive to both the convergence and the diversity of solutions.

The Mutual Domination Rate (MDR) measures the relative progress between two consecutive generations by comparing the dominance relationships between Pareto front approximations (Audet et al. 2021):

$$MDR(Y_N; n) = \frac{|\Delta(Y_N(n-1), Y_N(n))|}{|Y_N(n-1)|} - \frac{|\Delta(Y_N(n), Y_N(n-1))|}{|Y_N(n)|} \tag{2}$$

**where:**

- $Y_N(n)$: Pareto front approximation at generation $n$
- $\Delta(A, B)$: Set of elements in $A$ that are dominated by at least one element of $B$
- $|\cdot|$: Cardinality (number of elements in the set)

The values of the MDR metric can be between -1 and 1 where MDR = 1 corresponds to generation n completely dominates generation n-1, MDR = -1, generation n is completely dominated by n-1 generation and MDR = 0 means that there is no progress between generation n and n-1 (Audet et al. 2021). Other convergence metrics exist, including the Generational Distance (GD), which measures the average distance between the approximated solutions and the reference Pareto front and the R2 indicator, which shares similarities with the hypervolume (Audet et al. 2021).

Diversity metrics evaluate the extent to which solutions are well distributed and uniformly cover the Pareto front, which is essential for providing a decision-maker with a representative set of trade-offs. Entropy assesses the difference between the distribution of solutions on two successive Pareto fronts (Saxena et al. 2016), based on the Kullback-Leibler divergence:

$$S(p\|q) = KL(p\|q) = - \sum_{i=1}^{T} p(x_i) log \frac{q(x_i)}{p(x_i)} \tag{3}$$

**where:**

- $KL$ : relative entropy for comparing two different distribution (also known as Kullback-Leibler divergence)
- $p(x_i)$ : probability distribution of Pareto front P
- $q(x_i)$ : probability distribution of Pareto front Q

When this measure approaches zero or stabilizes, it indicates that the distribution of solutions no longer evolves significantly between generations, which constitutes a signal that the diversity of the front has stabilized. Other diversity metrics are also employed, such as the Spacing, which measures the uniformity of the solution distribution and the Spread, which quantifies the extent of the approximated front.

By monitoring the evolution of these metrics across generations, stopping criteria can be constructed and triggered when a given metric stagnates or crosses a predefined threshold, indicating that the algorithm no longer produces significant improvement and that solution quality has stabilized.

## 2.2 Formulation of stopping criteria

### 2.2.1 Overview of the existing stopping criteria

In this section, the four main stopping criteria introduced in the literature are recalled: OCD HV, LSSC, MGBM and Entropy. Three of them are based on a convergence metric and one on a diversity metric.

**OCD HV** and **LSSC** criteria are based on hypervolume (Guerreiro et al. 2022) (1). The OCD HV criterion seeks to determine if the improvement in hypervolume is statistically significant over a window of generations with a Chi-squared test (p-value < 0.05). The LSSC criterion is based on the slope and dispersion of the residuals from a linear regression. It requires the size of the window to be fixed as well as a threshold on the slope. In this article, the threshold is set at $10^{-2}$ according to the literature.

**MGBM** criterion is based on the MDR metric (2) which measures the number of non-dominated solutions from the previous iteration that are dominated by non-dominated points in the current iteration. The stopping criterion is configured to terminate the iteration when the MDR drops below a fixed threshold (here, 0.05), indicating that the front is no longer progressing (Abu Doush et al. 2023).

The **Entropy** criterion evaluates the diversity of solutions along the Pareto front using the Kullback-Leibler divergence (3). The measure relies on rounded means over a window of generations, with the rounding precision being user-defined. According to the literature, this is typically set to two decimal places (Saxena et al. 2016).

For all the criteria requiring a sliding window of generations to assess the stopping condition, the size of this window was set to 10 generations.

### 2.2.2 Definition of MPF criterion

The MPF stopping criterion is based on the previously introduced evaluation metrics of hypervolume (1) and entropy (3), which assess the convergence and the diversity of the Pareto front approximation, respectively. While hypervolume alone captures the overall quality of the front, it does not distinguish between a loss of convergence and a loss of diversity. Incorporating entropy as a complementary condition ensures that the algorithm terminates only when both the coverage and the distribution of solutions have stabilized. The full MPF criterion procedure is detailed in Algorithm 1 (Appendix).

At each generation, the hypervolume and entropy values are recorded. For each sliding windows of $L$ generations, the mean entropy and the relative variation in mean hypervolume $\Delta H$ are computed. The algorithm terminates when two conditions are simultaneously satisfied: the relative variation in mean hypervolume between the two windows, $\Delta H$, falls below a threshold $\varepsilon_H$ and the rounded mean entropy values of both windows are equal with a defined rounding precision $\varepsilon_S$, indicating that the solution distribution is no longer evolving. This joint condition prevents premature stopping in situations where the front has converged geometrically but the distribution of solutions remains unstable or conversely, where entropy has stabilized but the hypervolume is still improving.

Since hypervolume values are inherently scale-dependent and grow with the number of objectives, the relative variation $\Delta H$ (Algorithm 1, line 21) is used instead of an absolute difference, making the threshold $\varepsilon_H$ comparable across problems of different dimensionalities. In this paper, $\varepsilon_H$ is set to $10^{-3}$, the window length $L$ to 10 generations and the rounding precision $\varepsilon_S$ for entropy to two decimal places, consistent with the values used for the other criteria evaluated in this study.

# 3 Methodology

Three complementary approaches were implemented in RStudio (version 2025.09.2) to compare the different stopping criteria. The first approach involves testing the criteria on a simulation of objective functions in order to generate a theoretical Pareto front. This makes it possible to compare the fronts obtained using the stopping criteria with the simulated front. The second situation evaluates the criteria using a benchmark of well-known multi-objective problems from the literature. The third situation compares the criteria using one selected industrial case. The following sections will detail these three comparative analyses.

A unified methodology was applied across all analyses to enable systematic comparison of the stopping criteria. For each optimization run, the maximum number of generations was set to 5,000. The stopping criteria are then applied and the following informations are collected for each criterion:

- Number of generations at the time of termination
- Hypervolume (HV)
- Spread
- Overall computation time (s)
- Criterion computation time (s)

The hypervolume is used to assess the convergence of the front obtained for each criterion, while the spread metric gives an indication of the diversity of the front. Two computational time metrics are also recorded. The overall computation time represents the complete execution duration from the first generation to the stopping generation and reflects the total cost of the optimization process. The criterion computation time corresponds to the time required to evaluate the stopping criterion at each generation. Although this cost may be negligible for benchmark problems optimization, it becomes a relevant factor in industrial real-time optimization contexts, where decisions must be communicated rapidly to the operator in order to adjust the process accordingly.

The multi-objective evolutionary algorithm NSGA-II was used to solve multi-objective problems (Deb et al. 2002). To account for the inherent variability of stochastic population initialization, which can impact optimization results, we performed 100 independent runs for each problem, thus enabling a representative distribution of the results.

## 3.1 Simulation of a theoretical Pareto front

To generate a theoretical Pareto front, we simulate two continuous objective functions, $Y_1$ and $Y_2$, defined over a two-dimensional input space $(X_1, X_2) \in [-10, 10]$. Each objective function is modeled

as a sum of weighted Gaussian functions, allowing precise control over the mean, standard deviations. Each objective $Y_k$ (for $k = 1, 2$) is computed as:

$$Y_k(X_1, X_2) = \sum_{i=1}^{n_k} h_{k,i} \cdot \exp\left(-\frac{(X_1 - \mu_{k,i}^{(1)})^2 + (X_2 - \mu_{k,i}^{(2)})^2}{2\sigma_{k,i}^2}\right) \tag{4}$$

**where:**

- $n_k$ is the number of Gaussian components for objective $Y_k$,
- $\mu_{k,i} = (\mu_{k,i}^{(1)}, \mu_{k,i}^{(2)})$ is the mean of the $i$-th Gaussian for objective $Y_k$,
- $\sigma_{k,i}$ is the standard deviation of the $i$-th Gaussian,
- $h_{k,i}$ is the weight of the $i$-th Gaussian.

The domain is discretized using a regular grid of size $50 \times 50$, forming a total of 2500 input points. This formulation produces smooth surfaces where the means, standard deviations and weights of the Gaussian functions influence the trade-offs between objectives. The theoretical Pareto front is obtained by evaluating all solutions in the objective space $(Y_1, Y_2)$ and selecting only the non-dominated points (those for which no other solution improves one objective without degrading the other). However, the evaluated grid does not capture all non-dominated points but only a subset, providing an approximation of the theoretical Pareto front. The simulated objectives $Y_1$ and $Y_2$, along with the corresponding approximate Pareto Front, are shown in Figure 1.

## 3.2 Multi-objective problem benchmarks

To compare the selected stopping criteria, several multi-objective optimization benchmark problems were selected, covering a variety of Pareto front structures. The well-known and widely used ZDT1 problem has a convex and continuous front, making it a simple case for an initial comparison (Deb et al. 2005). The WFG2 problem introduces complexities with a non-convex and discontinuous front, as well as multimodality in the form of plateaus. A problem is multimodal if it contains several local optima (solutions not locally dominated but dominated by other solutions on the front). The WFG3 problem exhibits degeneracy due to the high correlation between objectives, providing a good case study for assessing the robustness of the criterion in relation to redundancy and dependency between objectives (which is common in real problems). Finally, WFG4, characterized by high multimodality and a concave front, is useful for testing the criterion's ability to avoid local minima (Huband et al. 2006). The large scope of these benchmark scenarios makes it possible to compare the performance of the stopping criteria on fronts of various shapes and complexities. In addition, as the WFG problems are scalable, they were used to analyze the impact of increasing the number of objectives on the behavior of the stopping criterion, with tests carried out for 2 objectives, then 4 and 8 objectives with 20 variables for both configurations. Problem ZDT1 was used in its classic configuration, with 2 objectives and 30 variables. Although the full ZDT and WFG suites were not used, the aim was to sample a variety of Pareto front shapes (such as convex, concave, discontinuous, degenerate and multimodal) in order to draw meaningful conclusions about the behavior of the stopping criteria across different problem types.

## 3.3 Industrial application

To test the stopping criteria on an industrial application, we analysed a case study from a cheese industry. This industry collects data throughout the manufacturing process, from milk collection to cheese ripening. These data were used for a multi-objective optimization coupled with data-driven modeling (Perrignon et al. 2024).

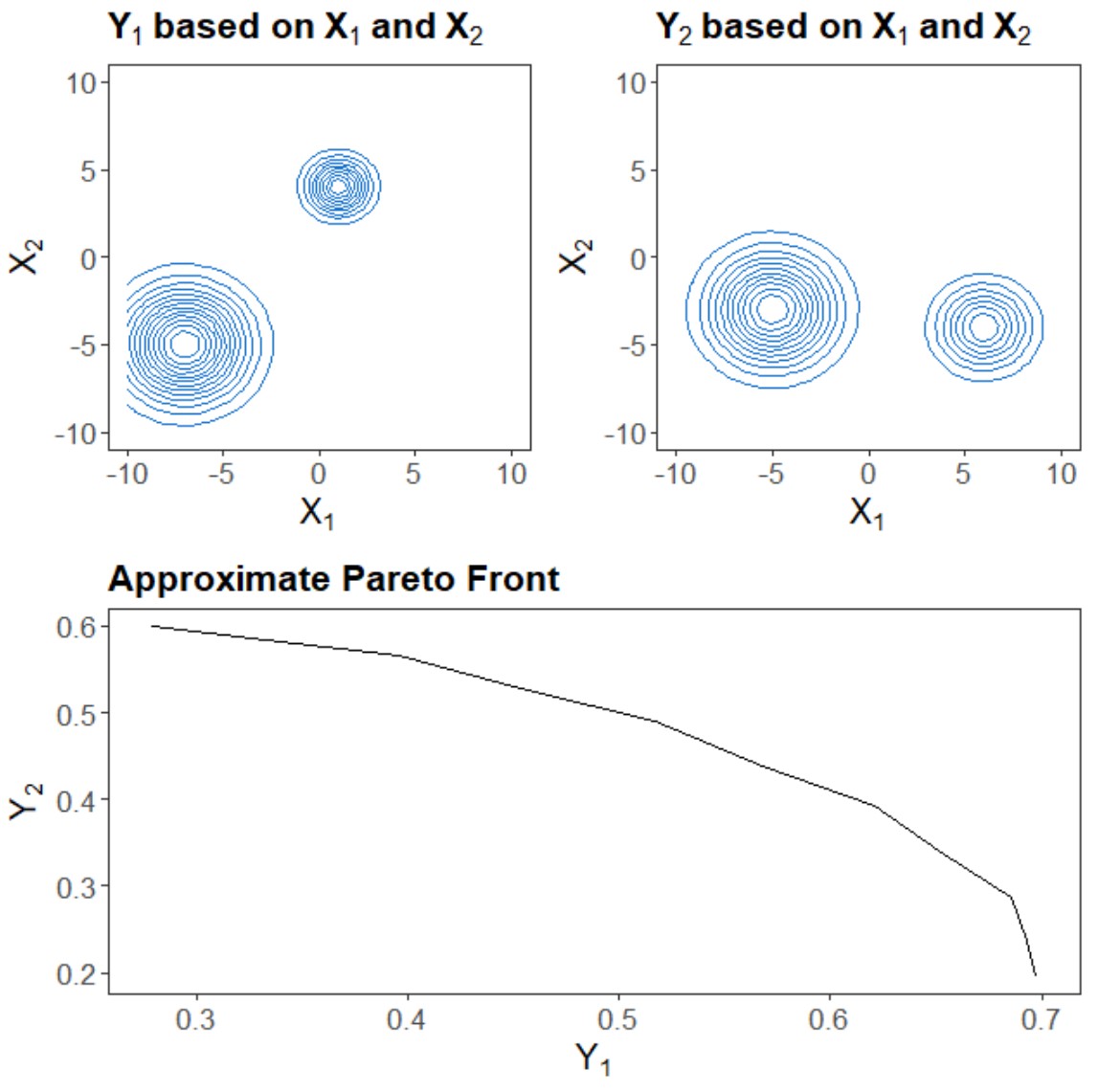

Figure 1: Visualization of $Y_1$ and $Y_2$ as a function of $X_1$ and $X_2$ and the approximate Pareto front obtained from the functions of $Y_1$ and $Y_2$. The aim is to maximize $Y_1$ and $Y_2$.

This approach involves defining the industry's performance objectives. Based on expert knowledge, four objectives were identified, each corresponding to a key performance indicator measured on the process. These indicators include economic performance indicators (cheese dry matter, protein content per kilogram of cheese and fat content per kilogram of cheese), as well as environmental indicator (water consumption).

Each indicator was modeled using Random Forest based on 79 process variables (Perrignon et al. 2025). These models are then used to construct the objective functions, which returns predicted values for the four indicators based on the input variables generated by the multi-objective optimization algorithm. The multi-objective problem can therefore be defined as follows:

$$\min \hat{f}_k(x)$$

$$\text{subject to } g(x) \geq \mathbf{0}$$

**with:**

$$
\begin{aligned}
&x \in \mathbb{R}^n && \text{(n process variables)} \\
&g \, : \, \mathbb{R}^n \to \mathbb{R}^j && \text{(j constraints)} \\
&f \, : \, \mathbb{R}^n \to \mathbb{R}^k && \text{(k objective functions)}
\end{aligned}
$$

$\hat{f}_k(x)$ is a machine learning model corresponding to indicator. The constraints of the optimization problem reflect the operational constraints of the cheese manufacturing process. These constraints apply to the process variables and may be expressed either as bounds or as equations linking some variables within the process.

## 4 Experimental results

### 4.1 Comparison with the simulated Pareto front

The stopping criteria were evaluated on optimization tasks using the simulated objective functions. Figure 2 reports the results over 100 repetitions for each criterion and Figure 3 shows the fronts obtained after the iteration is stop by the stopping criteria compared with the simulated front. Detailed numerical results are provided in the appendix (Table 1).

According to Figure 2, the differences between the stopping criteria are not particularly pronounced in terms of hypervolume values. All criteria appear to successfully reach the Pareto front (Figure 3). However, there are variations in the number of generations before termination. The Entropy criterion tends to stop after a higher number of generations than the OCD HV or LSSC criteria. Since the objective functions (i.e. $Y_1$ and $Y_2$ as functions of $X_1$ and $X_2$) are relatively simple, the Pareto front stabilizes after only a few generations. This stabilization is accurately detected by all criteria.

As expected, OCD HV stops before the other criteria, resulting in lower computational cost and potentially reduced performance in terms of hypervolume values. The Pareto front obtained under OCD HV is satisfactory, although some areas of the front remain unexplored (Figure 3). On the other hand, Entropy stops much later, leading to higher-quality fronts at the expense of increased computational cost. The Entropy criterion produces results similar to those of the MGBM and MPF, but requires more generations to do so. This suggests that Entropy could potentially terminate earlier while still achieving comparable Pareto front quality. The LSSC criterion obtains similar results to

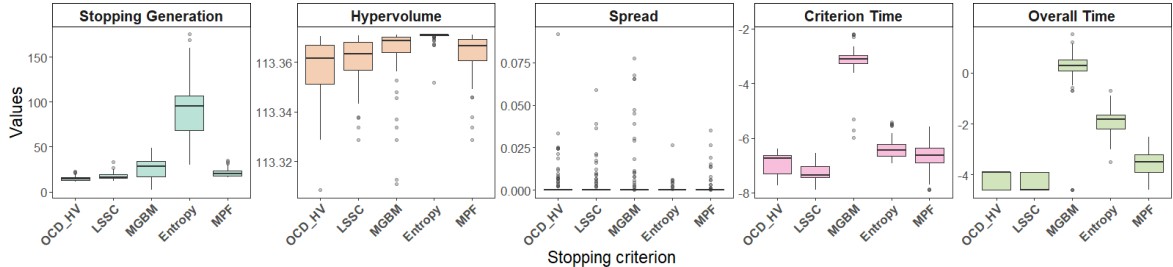

Figure 2: Optimization results for a simulated problem across stopping criteria. For each criterion, 100 independent optimization runs were performed. The figure reports the number of generations, the hypervolume (higher is better), the spread (lower is better) and two measures of computation time (log-transformed).

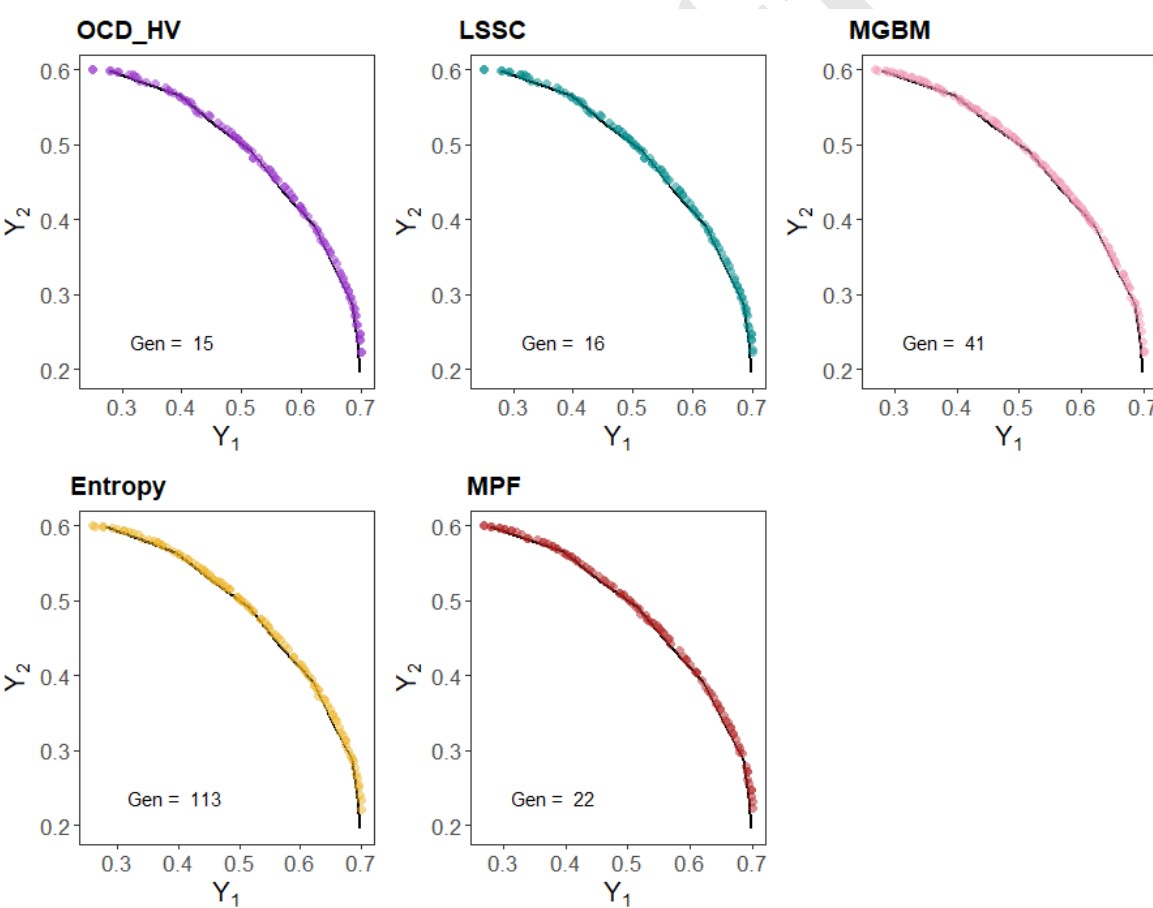

Figure 3: Comparison of the Pareto Fronts obtained for each stopping criterion with the simulated Pareto Front. The black line corresponds to the simulated Pareto front

OCD HV with some unexplored regions. For this simple problem, characterized by a small number of objectives and variables, all the stopping criteria identify a Pareto front close to the simulated front, with no significant differences in their ability to reach the theoretical front.

## 4.2 Increase in complexity with problems benchmark

### 4.2.1 Results for two-objective problems

This section focuses on analysing benchmark problems with two objectives. The four problems presented here represent a range of difficulty levels and exhibit diverse Pareto front shapes. Figure 4 presents the results in terms of the number of generations when the iteration is stop by the stopping criteria, the corresponding hypervolume and spread, as well as two measures of computational time (criterion time and overall time). Detailed numerical results are provided in the appendix (Table 2).

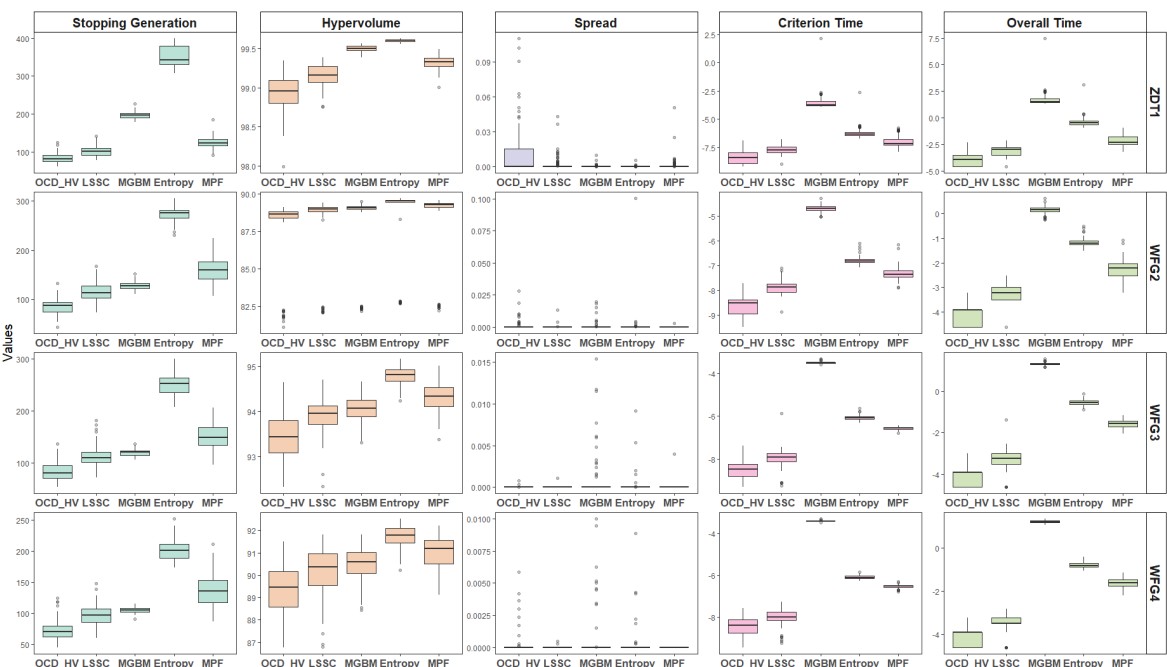

Figure 4: Optimization results for problems benchmark with k = 2. For each criterion, 100 independent optimization runs were performed. The figure reports the number of generations, the hypervolume (higher is better), the spread (lower is better) and two measures of computation time (log-transformed).

The stopping criteria behave consistently across the four two-objective problems and the overall trends agree with previous findings in the literature. The OCD HV criterion, designed to minimize computational effort (Wagner et al. 2009), stops earlier than the other methods. It therefore reaches lower hypervolume values but remains the fastest in terms of computation time. The LSSC criterion offers intermediate performance. It performs better than OCD HV for hypervolume and spread but remains less effective than the other criteria. The MGBM criterion reaches high hypervolume and low spread on all problems, although this comes with a substantial computational cost that appears in both the criterion time and the overall time. The Entropy criterion stands out as the most effective method. It achieves high hypervolume and low spread while maintaining a much lower computational cost, which makes it the most efficient option in this study. The MPF criterion provides a trade-off. It delivers satisfactory hypervolume and spread with low computation time, offering a reasonable balance between solution quality and efficiency.

Figure 5 illustrates the evolution of the hypervolume over generations for problems benchmark for

one run of optimization and the generation at which the stopping criteria stop the iteration. The stopping criteria identify the generation at which the Pareto front is considered stabilized, enabling the algorithm to terminate the iteration processus. To further illustrate the behavior of each stopping criterion, the obtained Pareto fronts are compared against the theoretical Pareto fronts of each benchmark problem in appendix (Figure 13).

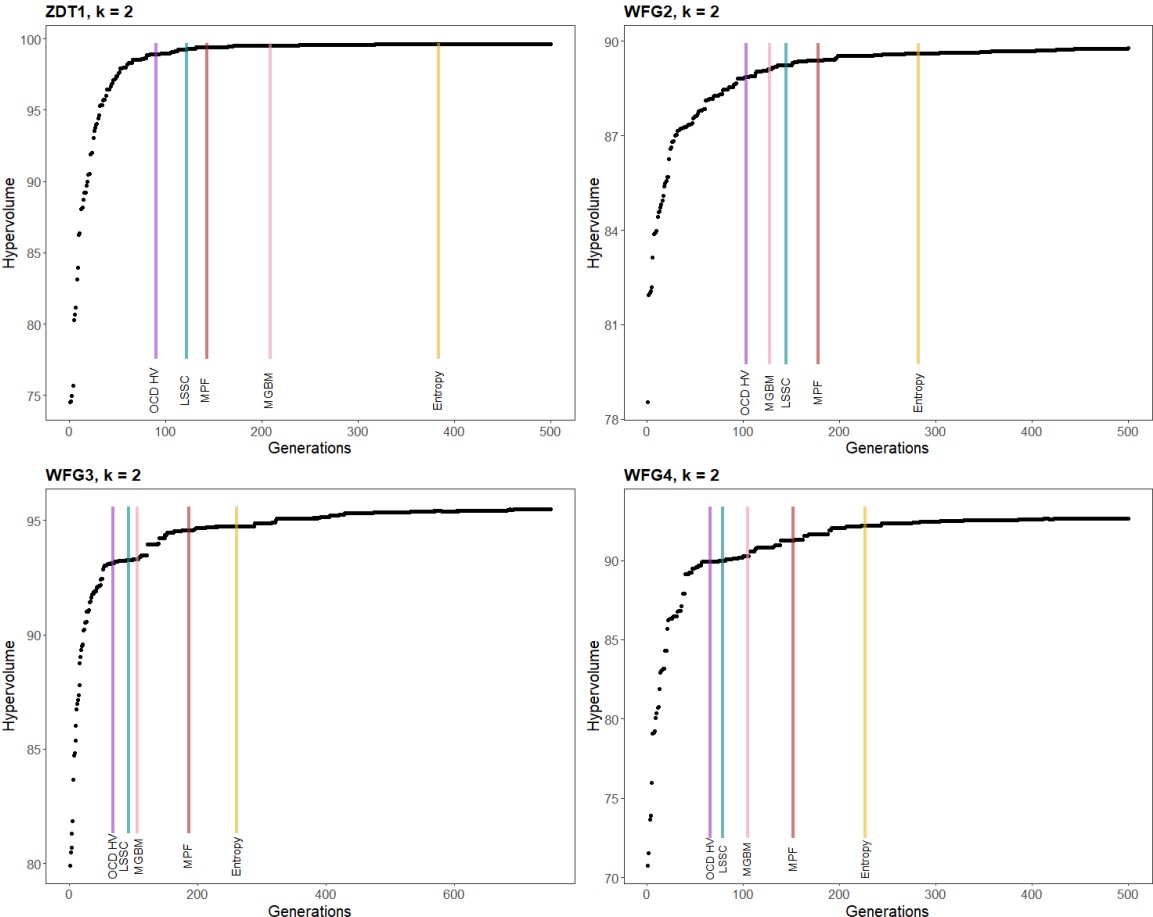

Figure 5: Evolution of the hypervolume over generations for the problems benchmark with the stopping generation proposed by each stopping criterion when k = 2.

For the ZDT1 problem, OCD HV and LSSC, are the first to stop, whereas the Pareto front is still undergoing slight improvements. The other criteria detect a later stabilization, especially MGBM and Entropy. MPF appears to offer the best compromise between computational cost and Pareto front quality. MGBM and Entropy stop much later, resulting in higher computational costs without improving the quality of the front, which is already stabilised.

For WFG2, which is characterized by a discontinuous Pareto front, the hypervolume curve shows a long and slow progression before reaching stability. OCD HV, MGBM and LSSC all stop while the curve is still improving, which means that they miss part of the progression of the front. MPF stops later, at a moment when the curve begins to stabilize, although a small improvement is still visible after its stopping point. Entropy stops well after the other criteria and does so when the front is clearly stable. It is the only criterion that fully captures the end of the improvement phase. As shown in the appendix figure, one fragment of the discontinuous front is not explored by any of the criteria. This confirms that the discontinuity of WFG2 makes the search more difficult and that all criteria fail to reach one part of the true Pareto front.

For the WFG3 problem, characterized by correlated objectives, the hypervolume increases in several steps. OCD HV, LSSC and MGBM stop on the first plateau and do not capture the later improvements of the curve. MPF and Entropy stop on a later plateau that gives a better hypervolume value, although a final small improvement is still visible after their stopping point and is not captured by any criterion. Entropy stops after MPF but both criteria reach the same hypervolume value. As in the other problems, Entropy often stops later than the other criteria, which leads to a higher computational cost.

Finally, for WFG4, OCD HV, LSSC and MGBM stop on the first plateau and do not capture the final stabilization of the front. WFG4 is characterized by multimodality, meaning the presence of multiple local minima. Both OCD HV, LSSC and MGBM become trapped at a local minimum and fail to escape it, preventing them from reaching the true optimality. MPF stops while the hypervolume is still improving and therefore misses the point where the curve becomes almost stable. Entropy is the only criterion that reaches the point where the front begin stable, which allows it to identify the optimal Pareto front.

In summary, for benchmark problems with two objectives, both OCD HV, LSSC and MGBM show comparable performance but do not always achieve front stability across different problems. MGBM also results in a long computation time. Entropy produces very strong results and consistently identifies high quality fronts. However, it often stops well after the front has already stabilized, which leads to unnecessary computation time. MPF achieves results that are very close to those of Entropy while stopping earlier. It therefore emerges as a promising choice for users who seek a good balance between solution quality and computational efficiency.

### 4.2.2  Results for four-objective problems

For WFG problems, it is possible to increase the number of objectives. To understand how the criteria respond to a higher number of objectives, we ran the WFG problems with four objectives. The results are shown in Figure 6 and detailed numerical results are provided in the appendix (Table 3).

The stopping criteria do not behave uniformly across the three WFG problems with four objectives. OCD HV fails to detect convergence on all three problems, reaching the maximum limit of 5,000 generations without identifying any stability in the Pareto front. A similar situation occurs with LSSC on WFG2 and WFG4, where no boxplot is visible, indicating that the criterion never triggered before the generation limit was reached. These two criteria therefore cannot be considered reliable stopping rules for these problems.

MGBM produces hypervolume values comparable to the other criteria on WFG2, but its overall computation time is substantially higher than all other criteria across all three problems. For WFG3 and WFG4, MGBM stops after very few generations, resulting in Pareto fronts of poor quality, as reflected by the low hypervolume values and high spread. This premature stopping behavior suggests that MGBM is sensitive to the complexity of the fitness landscape and does not scale well with problem difficulty, making it a computationally expensive and unreliable option for these problems.

Entropy and MPF show comparable hypervolume performance across all three problems and exhibit similar criterion computation times per generation. Among the two, Entropy consistently yields slightly lower hypervolume values than MPF, while both maintain low and stable computational costs. A notable observation concerns MPF, where the stopping generation boxplot is wide while the hypervolume boxplot remains narrow. This can be explained by the fact that, for certain optimization runs, MPF detects stability only at a later stage, causing it to stop at substantially higher generations. Yet, despite this delayed detection, the resulting front quality is comparable to that obtained at lower generations, indicating that the additional generations executed in these runs do not translate into a hypervolume gain. This combination of variable stopping generation and stable solution quality

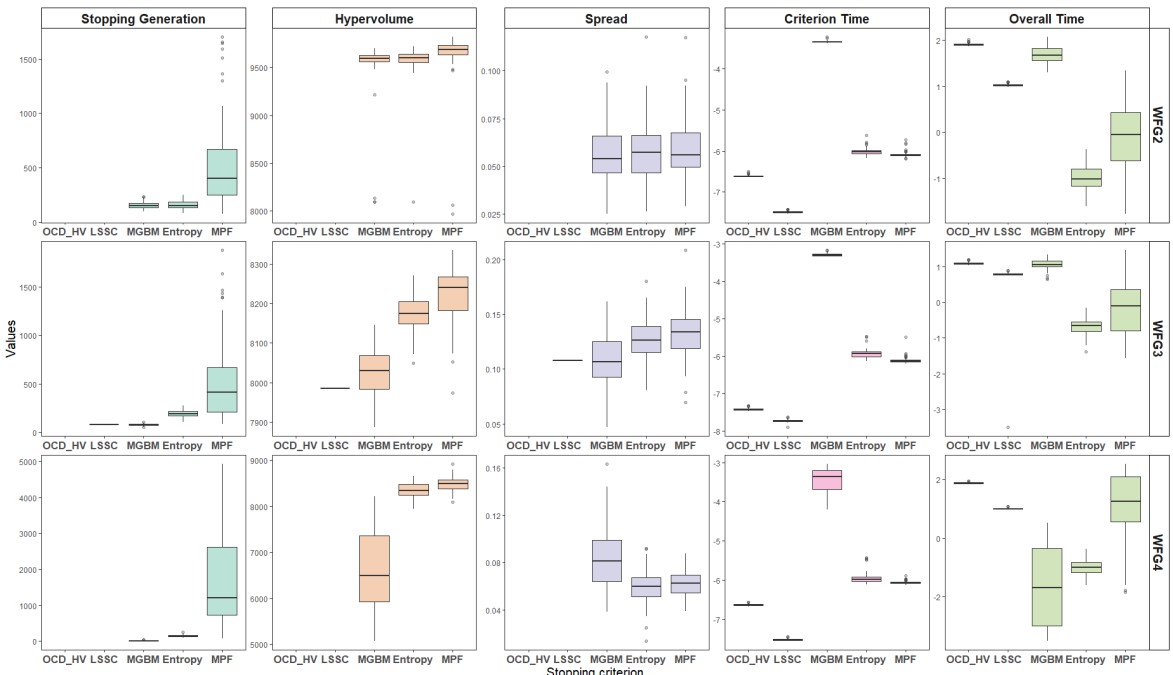

Figure 6: Optimization results for problems benchmark with k = 4. For each criterion, 100 independent optimization runs were performed. The figure reports the number of generations, the hypervolume (higher is better), the spread (lower is better) and two measures of computation time (log-transformed).

nonetheless comes at the cost of higher computational overhead, since the criterion sometimes requires substantially more generations to trigger without a corresponding improvement in front quality.

Figure 7 illustrates the evolution of the hypervolume across generations iteration and the generation the iterations stop for each criterion on the WFG problems.

Unlike the two-objective case, hypervolume evolution with four objectives becomes unstable and subject to fluctuations, which makes it more difficult for the stopping criteria to operate effectively. OCD HV and LSSC fail to detect stable convergence across all three problems, reaching the maximum limit of 5,000 generations and therefore are not represented in the figure.

On WFG2, MGBM and Entropy stop at a similar generation, shortly after the hypervolume plateau is reached. However, MPF stops considerably later, around generation 550, well after the plateau has been established. Despite this later stopping point, MPF still achieves a good trade-off between solution quality and computational cost.

On WFG3, the three criteria stop at different points along the hypervolume curve. MGBM stops the earliest, before the plateau is fully reached, which may result in an incomplete front. Entropy stops closer to the plateau, while MPF stops the latest, once the plateau is clearly established. This ordering suggests that MPF applies a more conservative convergence detection strategy, which is beneficial for problems where the hypervolume stabilizes gradually.

WFG4 exhibits large and persistent fluctuations in hypervolume values throughout the entire optimization process, making convergence detection particularly challenging. Both MGBM and Entropy stop early, at a point where the hypervolume has not yet stabilised. This suggests their difficulty in handling highly multimodal problems with noisy hypervolume values. MPF, by contrast, stops only once the hypervolume fluctuations subside enough to allow a stable trend to emerge. While this

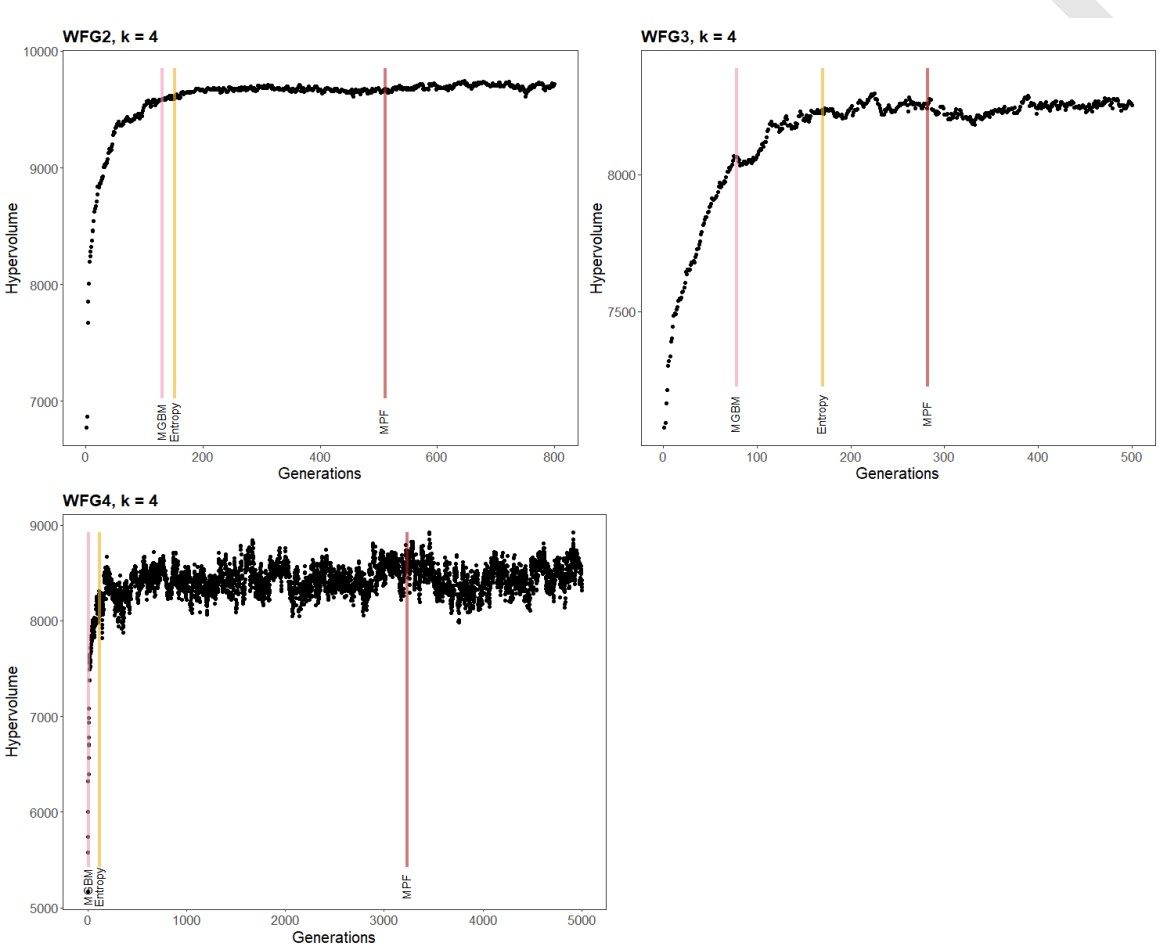

Figure 7: Evolution of hypervolume over generations for problems benchmark for k = 4 with the stopping generation proposed by each stopping criterion.

behavior leads to good quality Pareto fronts, it comes at the expense of a significantly higher computation time on this problem. The behavior of MPF is coherent with the wide stopping generation distribution observed in Figure 6 and reflects the inherent difficulty of detecting convergence on a highly multimodal problem.

In summary, increasing the number of objectives from two to four strongly impacts the behavior of the stopping criteria. OCD HV and LSSC become unable to detect Pareto front stability. MGBM and Entropy tend to stop prematurely on the more complex problems, sacrificing solution quality for early termination. For most problems, MPF consistently detects convergence and yields optimal-quality solutions even as problem complexity increases. However, for certain optimization runs, this detection occurs late, resulting in a higher computational cost.

### 4.2.3 Results for eight-objective problems

In order to assess the behavior of the criteria within a large number of objectives, problems WFG2, WFG3 and WFG4 were run with eight objectives. The results are shown in Figure 8 and detailed numerical results are provided in the appendix (Table 4).

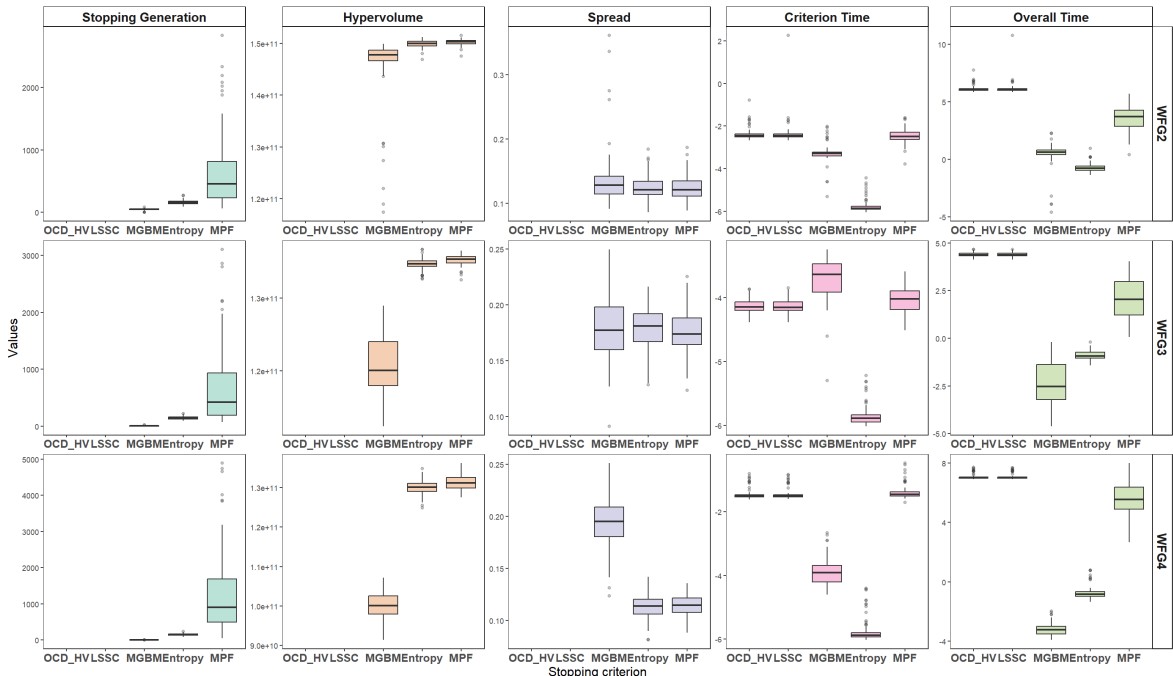

Figure 8: Optimization results for problems benchmark with k = 8. For each criterion, 100 independent optimization runs were performed. The figure reports the number of generations, the hypervolume (higher is better), the spread (lower is better) and two measures of computation time (log-transformed).

As shown in Figure 8, the behavior of the stopping criteria with eight objectives follows patterns consistent with those observed in the four-objective case, while amplifying some trends. OCD HV and LSSC fail to detect front stability across all three problems, confirming the lack of robustness of these two criteria as the number of objectives increases. This systematic failure suggests that both criteria are sensitive to the dimensionality of the objective space and cannot be considered reliable options for multi-objective optimization problems with a large number of objectives.

For the remaining three criteria, performance varies depending on the problem structure. On WFG2, Entropy and MPF achieve comparable hypervolume and spread values. MGBM, on the other

hand, yields slightly lower hypervolume values, with some runs producing particularly poor fronts, suggesting inconsistent convergence behavior.

On WFG3, Entropy and MPF outperform MGBM in terms of hypervolume. MGBM again incurs a substantially higher computational cost compared to the other two criteria, reinforcing the observation that its overhead is significant regardless of the problem complexity. MPF and Entropy produce similar results on this problem in terms of hypervolume and spread. However, MPF exhibits a considerably wider spread in its stopping generation, as reflected by the extent of its boxplot. This variability confirms the pattern already observed for four-objective problems, where MPF occasionally detects stabilization later than necessary, incurring unnecessary computational cost.

On WFG4, MGBM stops prematurely, yielding a significantly lower hypervolume than Entropy and MPF, which confirms its inability to handle highly multimodal problems regardless of the number of objectives. Entropy and MPF achieve similar hypervolume values but MPF requires a considerably higher number of generations before stopping. As observed in the four-objective case, this behavior can be attributed to the persistent hypervolume fluctuations characteristic of WFG4, which require MPF to wait until a stabilization emerges before stopping. While this leads to good quality fronts, it results in a substantially higher overall computation time for MPF compared to Entropy on this problem.

Across all problems, it is worth noting that the computational cost of hypervolume-based criteria (OCD HV, LSSC and MPF) is substantially higher than that of the Entropy criterion. The hypervolume indicator is well known for its exponentially increasing computational complexity as the number of objectives grows. This difference was not apparent for two or four objectives but becomes clearly observable at eight objectives, highlighting a scalability limitation of hypervolume-based stopping criteria in higher-dimensional settings.

To deepen the analysis, Figure 9 illustrates the evolution of the hypervolume across generations and the stopping point of each criterion on the WFG problems with eight objectives.

On WFG2, the hypervolume converges rapidly within the first 100 generations. The hypervolume trajectory exhibits persistent low-amplitude fluctuations throughout the remaining generations, a behavior that was not observed in the two and four-objective cases. Despite these fluctuations, the front quality remains stable. MGBM stops very early, before the hypervolume has fully stabilized, while Entropy detects convergence around generation 150, consistent with the plateau observed in the hypervolume evolution. MPF stops considerably later, around generation 800, suggesting that the persistent fluctuations delay the detection of a stable convergence signal. While a slight improvement in hypervolume is observed compared to Entropy, the gain remains marginal compared to the extra computational cost of the additional generations.

A similar pattern is observed on WFG3, where the hypervolume stabilizes rapidly but continues to exhibit fluctuations throughout the generations. MGBM again terminates prematurely, well before the front has stabilized. Entropy stops around generation 200, capturing the convergence accurately. MPF, however, stops around generation 800, again reflecting its sensitivity to hypervolume fluctuations, which prevent the relative variation from consistently falling below the threshold. This confirms that in higher-dimensional objective spaces, the hypervolume indicator becomes noisier, which penalizes criteria that rely on its stabilization.

On WFG4, the hypervolume rises steeply in the first hundred generations but never fully stabilizes, exhibiting large and persistent fluctuations across the entire run of 5000 generations. This high variability is characteristic of multimodal problems, where the algorithm continuously explores interesting regions of the objective space. This pattern prevent a smooth stabilization of the front. MGBM and Entropy both stop very early, around generation 50, at a point where the hypervolume

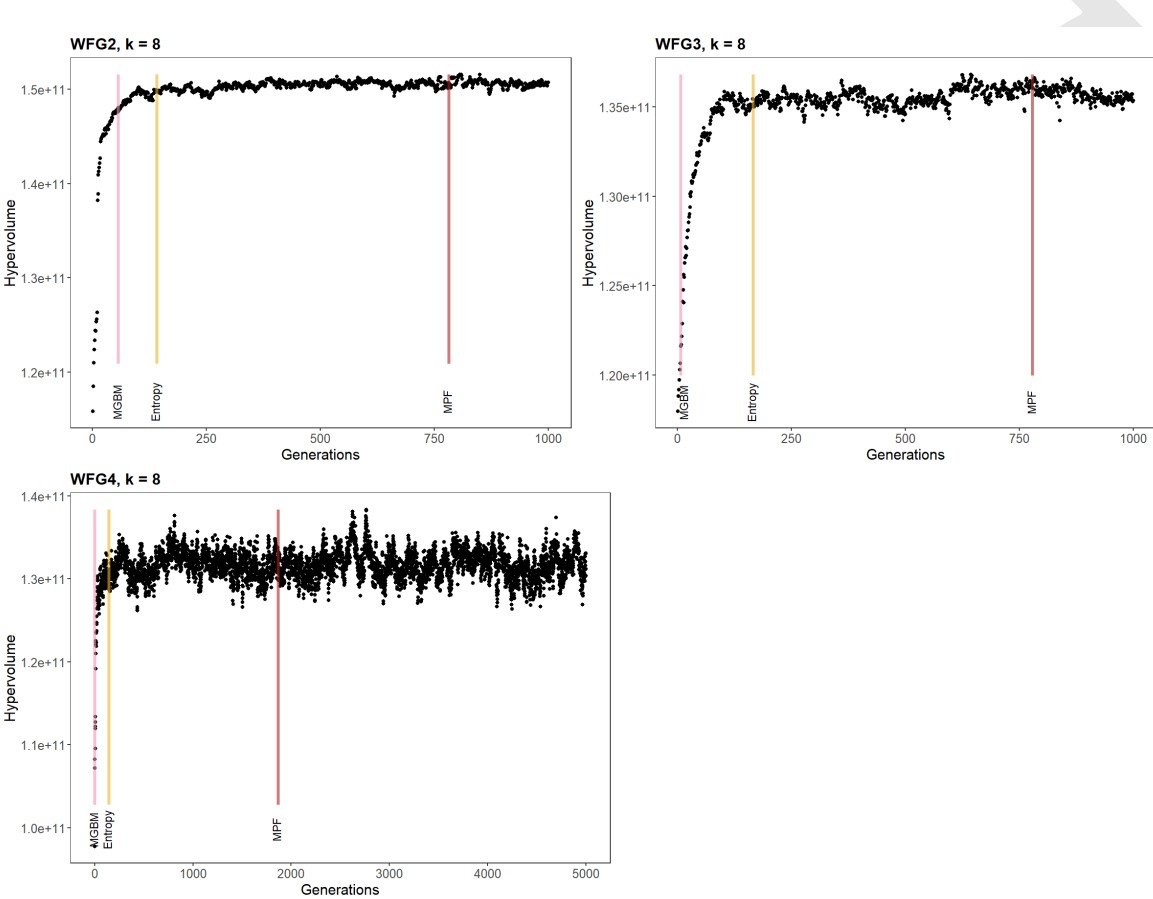

Figure 9: Evolution of hypervolume over generations for problems benchmark for k = 8 with the stopping generation proposed by each stopping criterion.

is still clearly increasing. This indicates premature termination and poor front quality. MPF stops much later, around generation 2000, which, while costly in terms of computation time, allows the algorithm to reach a higher hypervolume. The persistent fluctuations on WFG4 present a challenge for stopping criteria. They lead to premature termination for some criteria, while resulting in a higher computational cost for others.

In summary, the eight-objective results confirm and strengthen the conclusions drawn from the four-objective analysis. OCD HV and LSSC consistently fail as problem complexity increases. MGBM is still susceptible to premature stopping on complex problems and incurs high computational costs. Entropy and MPF emerge as the most robust criteria, with MPF providing better front stability at the cost of a higher computation time for the most challenging problems. However, it should be noted that hypervolume fluctuations were present across all three eight-objective problems, whereas these were either absent or negligible in the two and four-objective cases. This phenomenon can be attributed to the increased complexity of the objective space in higher dimensions, where the NSGA-II population is more sparsely distributed and small changes in the non-dominated set can produce relatively large variations in hypervolume. This increased noise in the hypervolume signal directly affects the behavior of stopping criteria, as illustrated by the results observed across the three problems discussed in this section.

## 4.3 Evaluation of the stopping criteria on a cheese-making process optimization problem

The stopping criteria were applied to the multi-objective optimization of a cheese production process. The problem was composed of four objectives and 79 variables and constraints linked to the process. Unlike the previous problems, this one has constraints that need to be integrated into the optimization. Figure 10 shows the performances of each criterion after optimization. Detailed numerical results are provided in the appendix (Table 5).

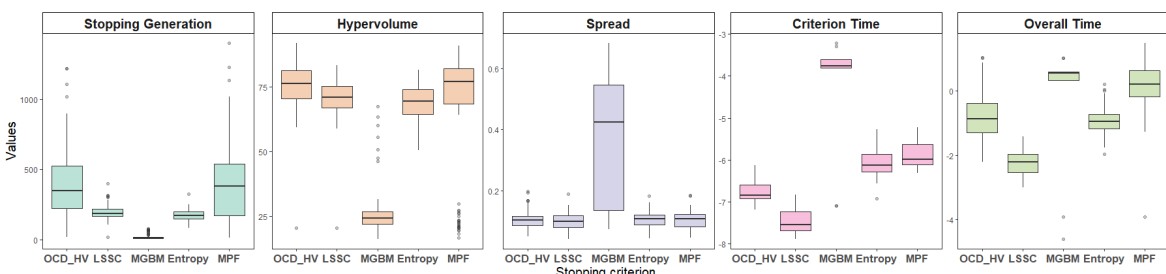

Figure 10: Stopping criteria for an industrial optimization with k = 4. The figure reports the number of generations, the hypervolume (higher is better), the spread (lower is better) and two measures of computation time (log-transformed).

MGBM is the first to stop, after only one generation. This premature termination logically results in a low hypervolume and high spread, which are typical of a front still in the exploratory phase. In comparison, LSSC and Entropy yield similar results, with Entropy requiring slightly more computation time. The OCD HV and MPF criteria obtain similar results in terms of Pareto front quality, with only a slight difference in computational time. However, both criteria show a high standard deviation, suggesting variability across runs and potential instability. MPF achieves good results despite this variability, making it a reasonable alternative to OCD HV with comparable performance.

To deepen the analysis, the evolution of the hypervolume over generations is illustrated in Figure 11, where the stopping generation for each criterion is indicated in color.

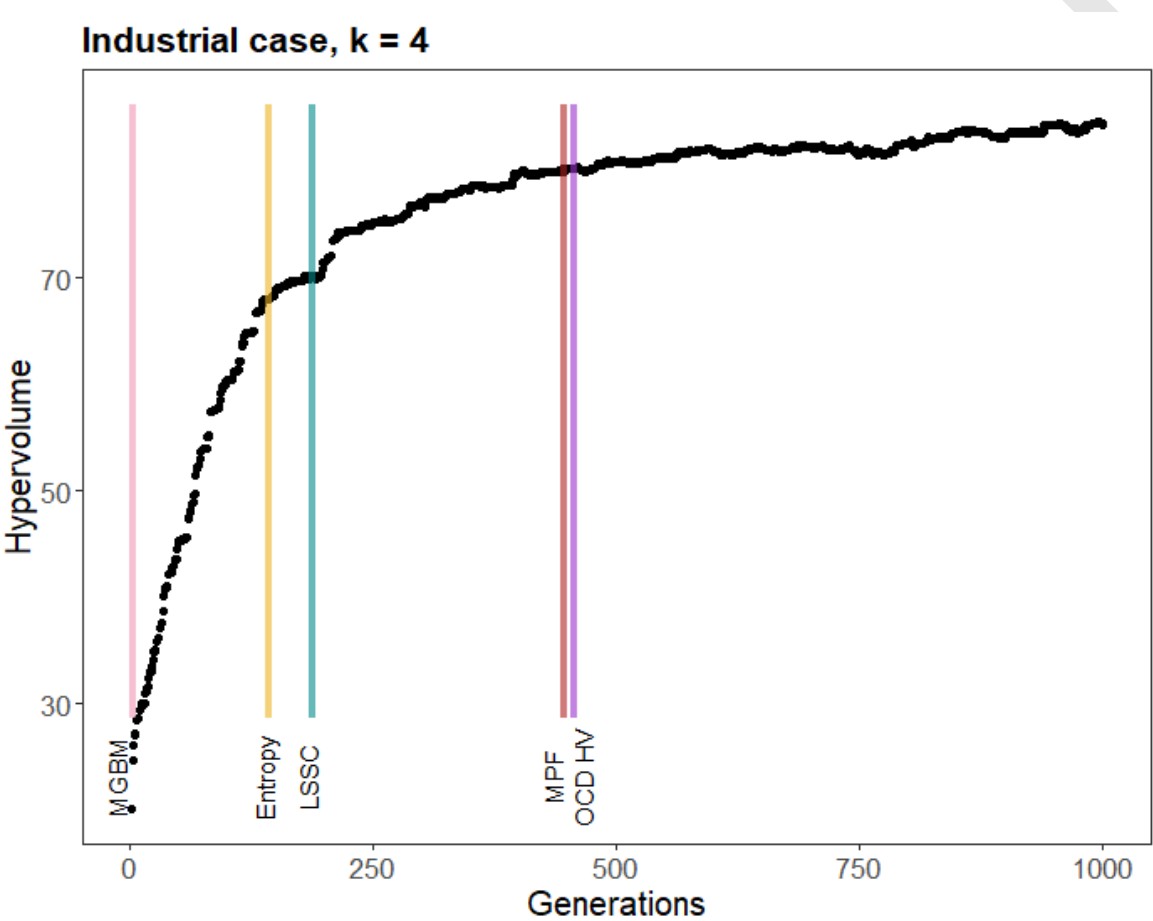

Figure 11: Evolution of hypervolume over generations for industrial optimization for k = 4 with the stopping generation proposed by each stopping criterion.

This curved evolution underlines an irregular hypervolume progression, characterized by plateaus, jumps and slight fluctuations. Such a trajectory is reminiscent of that observed in certain problems from the WFG family, which are known for their complexity. Several factors may explain this behavior: the presence of local optima, discontinuities in the Pareto front or correlations between objectives. Altogether, these elements suggest that the industrial problem under consideration is complex and difficult to optimize. In this context, MGBM stops too early, well before the Pareto front reaches a satisfactory quality. LSSC and Entropy also stop while the hypervolume is still increasing, indicating that they fail to reach an optimal front. OCD HV and MPF criteria demonstrate similar behavior, both stopping when the hypervolume starts stabilizing, indicating good Pareto front quality. Both criteria appear to effectively capture the convergence of the optimization process by terminating when the hypervolume becomes nearly stationary, reflecting a stabilization of the Pareto front.

In conclusion, in a real industrial context, most criteria face challenges similar to those observed in certain WFG problems, due to the complexity of the Pareto front. Among all the evaluated criteria, OCD HV and MPF stand out, as they are the only ones to propose stopping points that align with the stabilization of the hypervolume, making them reliable choices in real complex scenarios. Furthermore, despite their robustness, they maintain reasonable computational costs, which makes them particularly suitable for industrial applications where a balance between performance and efficiency is required.

# 5 Discussion and future work

## 5.1 Comparison of different criteria

A good stopping criteria implemented to multi-objective optimization problem algorithm should manage trading off between computational cost, robustness to problem complexity and the ability to detect algorithm convergence. MPF emerges as the most robust criterion across the problems evaluated, successfully combining high-quality Pareto fronts, computation times compatible with industrial constraints and resilience to varying levels of problem complexity. It is the only criterion capable of reliably detecting convergence in complex and realistic optimization settings, making it well-suited for industrial applications. However, as the number of objectives increases, MPF consistently identifies stabilization but requires a higher number of generations to do so, resulting in increased computational cost. In contrast, OCD HV and LSSC criteria, while having low computational cost for problems with few objectives and being simple to implement, see their effectiveness decreasing as the problem becomes more complex, particularly when the number of objectives increases or when the Pareto front has specific challenging features. In such cases, they fail to detect front stability accurately. MGBM can identify good quality Pareto fronts on problems of moderate complexity, but it is prone to premature stopping on the most challenging problems, such as WFG4 regardless of the number of objectives, leading to incomplete and suboptimal fronts. Furthermore, its computational cost is higher than all other criteria across all tested configurations, which limits its suitability in industrial contexts where rapid decision-making is required. Entropy, on the other hand, achieves very good results regardless of the problem complexity, while remaining computationally inexpensive. However, since it relies on a single metric, it may struggle in situations where both convergence and diversity analysis are necessary to fully assess the quality of the front. This limitation is highlighted in the real-world application case, where Entropy fails to capture the full diversity and convergence of the Pareto front.

## 5.2 Sensitivity of criteria to configuration parameters

In this study, five stopping criteria were compared, each designed with its own approach. This diversity in construction largely explains the differences in behavior observed when facing problems

of varying complexity. As the number of objectives and problem complexity increase, the LSSC and OCD HV criteria tend to fail in detecting stability reliably. This instability is mainly due to their convergence detection mechanism, which is based on statistical tests configured with a sliding window and a significance threshold. When the metric values fluctuate strongly, as observed in WFG4, these parameters become insufficient to detect a clear trend toward convergence. Adjusting the window size (e.g., by increasing it) or modifying the significance threshold could help smooth out these variations and better identify front stability. The choice of threshold is, in fact, a critical parameter for all criteria. A study on the Entropy criterion showed that the number of decimal places used for the metric can significantly influence the stopping point: the higher the required precision (e.g., rounding to 3 or 4 decimal places), the greater the number of generations needed (Saxena et al. 2016).

Regarding MGBM, a fixed threshold of 0.05 was used (Abu Doush et al. 2023). This threshold is intended to detect convergence when the MDR value approaches zero, indicating absence of significant improvements. However, our results show that MDR values vary significantly depending on the number of objectives for WFG4. The behaviour of the MDR value differs substantially between two, four and eight-objective problems. For two-objective problems, the 0.05 threshold is reached only after a huge number of generations, reflecting a true stabilization in the evolutionary process and indicating real convergence. In contrast, for four and eight-objective problems, the same threshold is reached almost immediately, often within the first few generations and therefore does not reflect real convergence or plateau. Conversely, a stricter threshold (such as 0.0001 (Martí et al. 2016)) is never reached in our experiments, preventing the detection of any stopping point. These observations highlight the limitations of using a static threshold for all problems. It may lead either to premature termination or to no termination at all. A more adaptive approach, based on the progression of MDR over generations rather than its value at one generation, would likely be more effective and better suited to specific problem characteristics.

The MPF criterion appears to be the most relevant across varying levels of problem complexity. For problems with two and four objectives, it offers a good compromise between Pareto front quality and computational cost. However, as the number of objectives increases, the hypervolume computation becomes increasingly expensive, which limits the practical appeal of MPF in higher-dimensional settings.

Many-objective optimization, typically defined as problems involving more than ten objectives, represents a challenging setting for stopping criteria. In our experiments, extending the evaluation to ten objectives revealed that hypervolume computation times become expensive, making it impractical to use as a basis for a stopping criterion in this context. This is a well-documented limitation in the literature, as the computational complexity of the hypervolume grows exponentially with the number of objectives (Guerreiro et al. 2022). It should be noted, however, that real-world engineering problems rarely involve ten objectives or even more and the two to four-objective case remains the most common in practice. Nevertheless, as optimization problems grow in complexity, the need for stopping criteria that remain computationally efficient in higher-dimensional spaces becomes increasingly relevant.

An important advantage of MPF in this regard is its modularity, as the hypervolume metric can be replaced by any other convergence metric. In many-objective optimization, computationally lower alternatives such as R2 or IGD could be substituted without altering the structure of the criterion (Audet et al. 2021; Liefooghe and Derbel 2016). This flexibility makes MPF particularly attractive for scaling to high-dimensional objective optimization problems, where computational efficiency becomes critical. The modular nature of MPF thus ensures its applicability across a wide range of problem complexities without being constrained by the computational time.

## 5.3 Future work

While the performance of the evaluated criteria was tested under increasing levels of problem complexity (such as objective correlation and multimodality), we did not explicitly consider another important form of complexity: a strong discontinuous nature of the Pareto front with convex and concave shapes, as exemplified by WFG1. This problem introduces a different challenge where the search process does not progress smoothly but rather by in steps or segments. WFG1 is characterized by a partially separable structure, transformation functions that create strong nonlinearities and a discontinuous Pareto front. As a result, optimization algorithms often discover the front piece by piece, trying to uncover disconnected regions. WFG2 also exhibits a discontinuous Pareto front but in a less complex form. While still posing difficulties, the segmentation in WFG2 is simpler and easier to overcome for the stopping criteria compared to WFG1's multi-fragmented structure. The tested criteria are not well-suited to capture this type of complexity as shown in Figure 12.

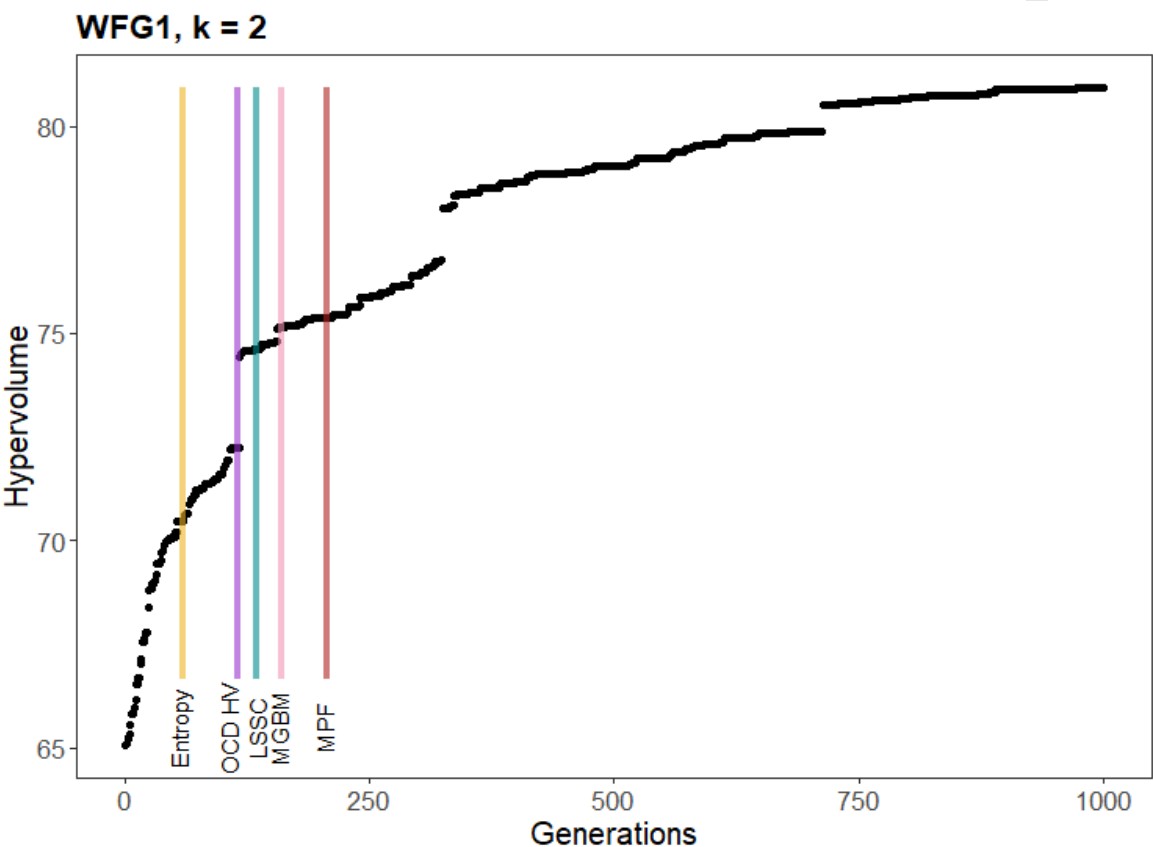

Figure 12: Evolution of hypervolume over generations for WFG1 problem for k = 2 with the stopping generation proposed by each stopping criterion.

The evolution of the hypervolume is discontinuous and the criteria stop at intermediate pillars, resulting in a non-complete exploration of the front. All criteria inherently assume a continuous and gradual improvement of the Pareto front, reflected in the smooth evolution of evaluation metrics like hypervolume. When this assumption does not hold, as in WFG1, these criteria may fail to detect meaningful progress or stability. This highlights an opportunity for improvement in scenarios where the front evolves with strong discontinuities. Future work presents opportunities to enhance these criteria by developing innovative mechanisms for managing front fragmentation, which will strengthen their effectiveness in complex problem scenarios.

More broadly, the benchmark selection in this study was guided by two main considerations: the need

to cover a range of Pareto front type (convex, concave, disconnected) and the ability to scale across different numbers of objectives. The WFG suite was preferred over a full ZDT evaluation because it satisfies both requirements simultaneously, whereas ZDT problems are restricted to two objectives and their front types are largely represented within the WFG suite. Nevertheless, extending the evaluation to a broader set of benchmark problems, including the remaining ZDT and WFG instances, remains an avenue for future work to further validate the general applicability of the conclusions drawn in this article.

# 6  Appendix

**Algorithm 1:** Stopping criterion using sliding windows on hypervolume and entropy

**Input:** $L$: window length, $\varepsilon_H$: convergence threshold, $\varepsilon_S$: number of decimal places

1. Initialize generation counter $n \leftarrow 0$
2. Initialize hypervolume history $H \leftarrow [\,]$, entropy history $S \leftarrow [\,]$
3. **while** true **do**
4.     Run NSGA-II for generation $n$ to obtain non-dominated set $Y_n$
5.     Compute hypervolume $HV_n \leftarrow HV(Y_n)$
6.     Compute diversity $Ent_n \leftarrow Entropy(Y_n, Y_{n-1})$
7.     Append $HV_n$ to $H$, and $Ent_n$ to $S$
8.     **if** $n \geq L + 1$ **then**
9.       Define previous window:
10.         $H_{\text{old}} \leftarrow H[n - L : n - 1]$
11.         $S_{\text{old}} \leftarrow S[n - L : n - 1]$
12.       Define current window:
13.         $H_{\text{new}} \leftarrow H[n - L + 1 : n]$
14.         $S_{\text{new}} \leftarrow S[n - L + 1 : n]$
15.       Compute mean values for both windows:
16.         $\overline{H}_{\text{old}} \leftarrow \frac{1}{L} \sum_{i=1}^{L} H_{\text{old}}[i]$
17.         $\overline{S}_{\text{old}} \leftarrow \frac{1}{L} \sum_{i=1}^{L} S_{\text{old}}[i]$
18.         $\overline{H}_{\text{new}} \leftarrow \frac{1}{L} \sum_{i=1}^{L} H_{\text{new}}[i]$
19.         $\overline{S}_{\text{new}} \leftarrow \frac{1}{L} \sum_{i=1}^{L} S_{\text{new}}[i]$
20.       Compute relative variations:
21.         $\Delta H \leftarrow \frac{|\overline{H}_{\text{new}} - \overline{H}_{\text{old}}|}{\overline{H}_{\text{old}}}$
22.       **if** $\Delta H \leq \varepsilon_H$ **and** $Round(\overline{S}_{\text{old}}, \epsilon_S) = Round(\overline{S}_{\text{new}}, \epsilon_S)$ **then**
23.         **break**
24.       **end if**
25.     **end if**
26.     $n \leftarrow n + 1$
27. **end while**
28. **return** non-dominated set $F_n$ and generation $n$

Table 1: Means and standard deviations of evaluation metrics for each criterion when k = 2 and X = 2.

| | Hypervolume | | Spread | | Generations | | Total Time (s) | | Criterion Time (s) | |
|---|---|---|---|---|---|---|---|---|---|---|
| Criterion | Mean | Std | Mean | Std | Mean | Std | Mean | Std | Mean | Std |
| OCD_HV | 113.358 | 0.011 | 0.004 | 0.011 | 15 | 2 | 0.003 | 0.007 | 0.000 | 0.000 |

| Criterion | | | | | | | | | | |
|---|---|---|---|---|---|---|---|---|---|---|
| LSSC | 113.361 | 0.009 | 0.003 | 0.009 | 17 | 4 | 0.005 | 0.007 | 0.000 | 0.000 |
| MGBM | 113.364 | 0.012 | 0.006 | 0.016 | 24 | 14 | 1.161 | 0.869 | 0.038 | 0.025 |
| Entropy | 113.371 | 0.002 | 0.001 | 0.003 | 93 | 30 | 0.163 | 0.078 | 0.002 | 0.001 |
| MPF | 113.363 | 0.008 | 0.002 | 0.005 | 21 | 4 | 0.030 | 0.018 | 0.001 | 0.001 |

Table 2: Means and standard deviations of evaluation metrics for each benchmark problem and criterion when k = 2.

| | | HV | | Spread | | Gen. | | Time (s) | | Crit. Time (s) | |
|---|---|---|---|---|---|---|---|---|---|---|---|
| Problem | Criterion | Mean | Std | Mean | Std | Mean | Std | Mean | Std | Mean | Std |
| ZDT1 | OCD_HV | 98.929 | 0.233 | 0.012 | 0.022 | 83 | 13 | 0.016 | 0.016 | 0.000 | 0.000 |
| | LSSC | 99.165 | 0.138 | 0.002 | 0.006 | 102 | 13 | 0.052 | 0.024 | 0.001 | 0.000 |
| | MGBM | 99.256 | 2.457 | 0.002 | 0.015 | 194 | 21 | 23.610 | 176.434 | 0.116 | 0.852 |
| | Entropy | 99.603 | 0.016 | 0.000 | 0.001 | 352 | 27 | 0.914 | 2.163 | 0.003 | 0.007 |
| | MPF | 99.322 | 0.082 | 0.001 | 0.006 | 125 | 14 | 0.136 | 0.087 | 0.001 | 0.001 |
| WFG2 | OCD_HV | 87.795 | 2.324 | 0.001 | 0.004 | 84 | 15 | 0.011 | 0.009 | 0.000 | 0.000 |
| | LSSC | 88.124 | 2.306 | 0.000 | 0.001 | 114 | 19 | 0.044 | 0.014 | 0.000 | 0.000 |
| | MGBM | 87.888 | 3.146 | 0.001 | 0.003 | 124 | 23 | 1.128 | 0.260 | 0.009 | 0.002 |
| | Entropy | 88.732 | 2.227 | 0.001 | 0.010 | 273 | 13 | 0.309 | 0.058 | 0.001 | 0.000 |
| | MPF | 88.404 | 2.318 | 0.000 | 0.000 | 162 | 27 | 0.112 | 0.046 | 0.001 | 0.000 |
| WFG3 | OCD_HV | 93.407 | 0.514 | 0.000 | 0.000 | 84 | 16 | 0.015 | 0.010 | 0.000 | 0.000 |
| | LSSC | 93.920 | 0.393 | 0.000 | 0.000 | 111 | 21 | 0.042 | 0.025 | 0.000 | 0.000 |
| | MGBM | 94.057 | 0.287 | 0.001 | 0.003 | 119 | 6 | 3.617 | 0.248 | 0.030 | 0.001 |
| | Entropy | 94.812 | 0.188 | 0.000 | 0.001 | 251 | 20 | 0.577 | 0.074 | 0.002 | 0.000 |
| | MPF | 94.320 | 0.302 | 0.000 | 0.000 | 151 | 24 | 0.211 | 0.040 | 0.001 | 0.000 |
| WFG4 | OCD_HV | 89.344 | 1.152 | 0.000 | 0.001 | 72 | 16 | 0.014 | 0.009 | 0.000 | 0.000 |
| | LSSC | 90.182 | 1.055 | 0.000 | 0.000 | 96 | 18 | 0.034 | 0.012 | 0.000 | 0.000 |
| | MGBM | 90.498 | 0.720 | 0.001 | 0.002 | 105 | 5 | 3.384 | 0.188 | 0.032 | 0.001 |
| | Entropy | 91.738 | 0.453 | 0.000 | 0.001 | 202 | 17 | 0.456 | 0.064 | 0.002 | 0.000 |
| | MPF | 91.051 | 0.724 | 0.000 | 0.000 | 138 | 26 | 0.203 | 0.042 | 0.001 | 0.000 |

Table 3: Means and standard deviations of evaluation metrics for each benchmark problem and criterion when k = 4. NA values indicate that the criterion reached the maximum number of generations (5000) without achieving front stability.

| | | HV | | Spread | | Gen. | | Time (s) | | Crit. Time (s) | |
|---|---|---|---|---|---|---|---|---|---|---|---|
| Problem | Criterion | Mean | Std | Mean | Std | Mean | Std | Mean | Std | Mean | Std |
| WFG2 | OCD_HV | NaN | NA | NaN | NA | NaN | NA | 6.72 | 0.16 | 0.001 | 0.000 |
| | LSSC | NaN | NA | NaN | NA | NaN | NA | 2.79 | 0.06 | 0.001 | 0.000 |
| | MGBM | 9549.86 | 261.89 | 0.06 | 0.02 | 155 | 30 | 5.52 | 1.05 | 0.036 | 0.001 |
| | Entropy | 9582.79 | 161.78 | 0.06 | 0.02 | 155 | 33 | 0.38 | 0.10 | 0.002 | 0.000 |
| | MPF | 9649.73 | 244.85 | 0.06 | 0.01 | 514 | 378 | 1.17 | 0.86 | 0.002 | 0.000 |
| WFG3 | OCD_HV | NaN | NA | NaN | NA | NaN | NA | 3.00 | 0.08 | 0.001 | 0.000 |
| | LSSC | 7985.19 | NA | 0.11 | NA | 81 | NA | 2.19 | 0.23 | 0.000 | 0.000 |
| | MGBM | 8026.45 | 54.15 | 0.11 | 0.02 | 78 | 10 | 2.91 | 0.35 | 0.037 | 0.001 |
| | Entropy | 8174.76 | 43.05 | 0.13 | 0.02 | 192 | 35 | 0.52 | 0.11 | 0.003 | 0.000 |
| | MPF | 8224.35 | 66.28 | 0.13 | 0.02 | 499 | 381 | 1.10 | 0.85 | 0.002 | 0.000 |

| Problem | Criterion | HV Mean | HV Std | Spread Mean | Spread Std | Gen. Mean | Gen. Std | Time (s) Mean | Time (s) Std | Crit. Time (s) Mean | Crit. Time (s) Std |
|---|---|---|---|---|---|---|---|---|---|---|---|
| | OCD_HV | NaN | NA | NaN | NA | NaN | NA | 6.57 | 0.15 | 0.001 | 0.000 |
| | LSSC | NaN | NA | NaN | NA | NaN | NA | 2.73 | 0.07 | 0.001 | 0.000 |
| WFG4 | MGBM | 6621.19 | 862.23 | 0.09 | 0.03 | 10 | 11 | 0.41 | 0.45 | 0.032 | 0.009 |
| | Entropy | 8351.38 | 158.02 | 0.06 | 0.01 | 147 | 31 | 0.39 | 0.11 | 0.003 | 0.000 |
| | MPF | 8483.86 | 140.97 | 0.06 | 0.01 | 1654 | 1256 | 5.10 | 3.95 | 0.002 | 0.000 |

Table 4: Means and standard deviations of evaluation metrics for each benchmark problem and criterion when k = 8. NA values indicate that the criterion reached the maximum number of generations (5000) without achieving front stability.

| Problem | Criterion | HV Mean | HV Std | Spread Mean | Spread Std | Gen. Mean | Gen. Std | Time (s) Mean | Time (s) Std | Crit. Time (s) Mean | Crit. Time (s) Std |
|---|---|---|---|---|---|---|---|---|---|---|---|
| | OCD_HV | NaN | NA | NaN | NA | NaN | NA | 483.78 | 219.47 | 0.097 | 0.044 |
| | LSSC | NaN | NA | NaN | NA | NaN | NA | 932.11 | 4760.66 | 0.186 | 0.952 |
| WFG2 | MGBM | 146106877647 | 6055653943 | 0.14 | 0.04 | 48 | 16 | 2.04 | 1.36 | 0.041 | 0.019 |
| | Entropy | 149897433459 | 695739758 | 0.12 | 0.02 | 159 | 35 | 0.52 | 0.28 | 0.003 | 0.001 |
| | MPF | 150250021210 | 549043680 | 0.12 | 0.02 | 606 | 562 | 54.76 | 52.04 | 0.088 | 0.029 |
| | OCD_HV | NaN | NA | NaN | NA | NaN | NA | 80.07 | 8.34 | 0.016 | 0.002 |
| | LSSC | NaN | NA | NaN | NA | NaN | NA | 79.99 | 8.35 | 0.016 | 0.002 |
| WFG3 | MGBM | 120906795330 | 4093040754 | 0.18 | 0.03 | 6 | 5 | 0.16 | 0.16 | 0.025 | 0.007 |
| | Entropy | 134765525373 | 758315444 | 0.18 | 0.02 | 145 | 29 | 0.42 | 0.11 | 0.003 | 0.001 |
| | MPF | 135300413211 | 719127712 | 0.18 | 0.02 | 682 | 681 | 12.87 | 13.18 | 0.018 | 0.004 |
| | OCD_HV | NaN | NA | NaN | NA | NaN | NA | 1176.60 | 233.40 | 0.235 | 0.047 |
| | LSSC | NaN | NA | NaN | NA | NaN | NA | 1176.76 | 237.46 | 0.235 | 0.047 |
| WFG4 | MGBM | 99997184717 | 3478706837 | 0.19 | 0.02 | 2 | 0 | 0.05 | 0.02 | 0.022 | 0.012 |
| | Entropy | 129992519846 | 1859657880 | 0.11 | 0.01 | 153 | 29 | 0.53 | 0.36 | 0.003 | 0.002 |
| | MPF | 131165928348 | 1771106201 | 0.11 | 0.01 | 1306 | 1145 | 414.35 | 455.02 | 0.253 | 0.070 |

Table 5: Means and standard deviations of evaluation metrics for each criterion for industrial case for k = 4.

| Criterion | Hypervolume Mean | Hypervolume Std | Spread Mean | Spread Std | Generations Mean | Generations Std | Total Time (s) Mean | Total Time (s) Std | Criterion Time (s) Mean | Criterion Time (s) Std |
|---|---|---|---|---|---|---|---|---|---|---|
| OCD_HV | 75.509 | 9.530 | 0.104 | 0.029 | 401 | 240 | 0.586 | 0.532 | 0.001 | 0.000 |
| LSSC | 70.407 | 7.538 | 0.099 | 0.028 | 192 | 51 | 0.113 | 0.044 | 0.001 | 0.000 |
| MGBM | 26.209 | 9.007 | 0.385 | 0.206 | 19 | 17 | 0.137 | 0.521 | 0.002 | 0.008 |
| Entropy | 69.009 | 6.652 | 0.104 | 0.025 | 174 | 37 | 0.427 | 0.189 | 0.002 | 0.001 |
| MPF | 66.441 | 23.941 | 0.105 | 0.028 | 396 | 308 | 1.155 | 0.997 | 0.002 | 0.001 |

# 7 Funding sources

This project was supported by Fondation Institut Agro

# 8 Authorship contribution statement

**Manon Perrignon:** Writing – original draft, Conceptualization, Methodology, Investigation, Simulation, Visualization. **Magalie Houée-Bigot:** Writing – review & editing, Methodology, Investigation, Simulation. **Romain Jeantet:** Writing – review & editing, Supervision, Conceptualization, Funding

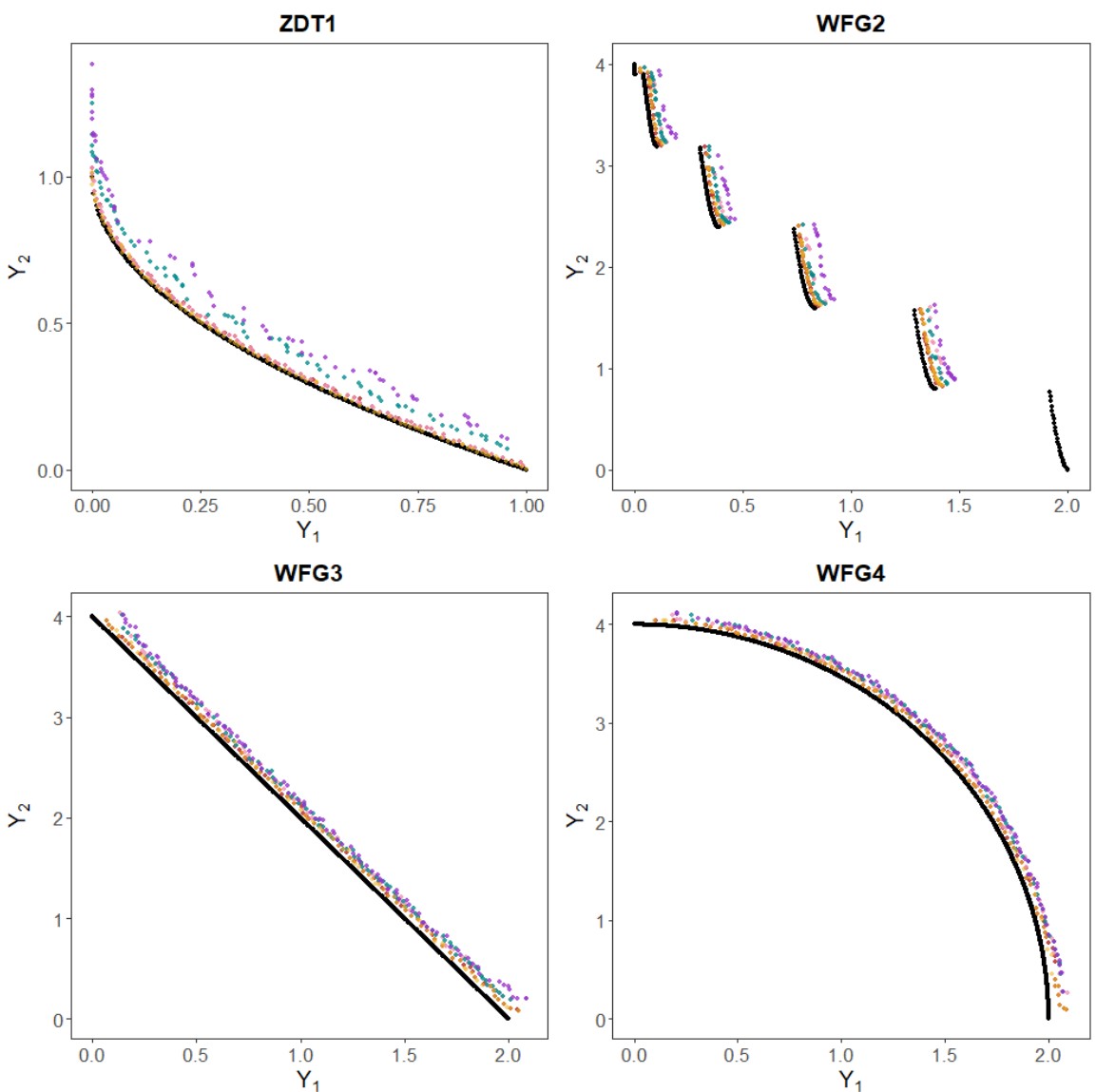

Figure 13: Comparison between the Pareto fronts obtained for each criterion and the theoretical Pareto fronts for the two objective benchmark problems.

acquisition. **Thomas Croguennec:** Writing – review & editing, Supervision, Conceptualization, Methodology. **Mathieu Emily:** Writing – review & editing, Supervision, Conceptualization, Methodology, Investigation.

Abu Doush, Iyad, Mohammed El-Abd, Abdelaziz I. Hammouri, and Mohammad Qasem Bataineh. 2023. "The Effect of Different Stopping Criteria on Multi-Objective Optimization Algorithms." *Neural Comput & Applic* 35 (2): 1125–55. https://doi.org/10.1007/s00521-021-05805-1.

Audet, Charles, Jean Bigeon, Dominique Cartier, Sébastien Le Digabel, and Ludovic Salomon. 2021. "Performance Indicators in Multiobjective Optimization." *European Journal of Operational Research* 292 (2): 397–422. https://doi.org/10.1016/j.ejor.2020.11.016.

Deb, Kalyanmoy, Lothar Thiele, Marco Laumanns, and Eckart Zitzler. 2005. "Scalable Test Problems for Evolutionary Multiobjective Optimization." In *Evolutionary Multiobjective Optimization: Theoretical Advances and Applications*, edited by Ajith Abraham, Lakhmi Jain, and Robert Goldberg. Springer. https://doi.org/10.1007/1-84628-137-7_6.

Deb, K., A. Pratap, S. Agarwal, and T. Meyarivan. 2002. "A Fast and Elitist Multiobjective Genetic Algorithm: NSGA-II." *IEEE Transactions on Evolutionary Computation* 6 (2): 182–97. https://doi.org/10.1109/4235.996017.

Drofenik, Jan, Bojan Pahor, Zdravko Kravanja, and Zorka Novak Pintarič. 2023. "Multi-Objective Scenario Optimization of the Food Supply Chain – Slovenian Case Study." *Computers & Chemical Engineering* 172 (April): 108197. https://doi.org/10.1016/j.compchemeng.2023.108197.

Ehrgott, Matthias. 2005. *Multicriteria Optimization*. Second edition. Springer.

Guerreiro, Andreia P., Carlos M. Fonseca, and Luís Paquete. 2022. "The Hypervolume Indicator: Problems and Algorithms." *ACM Comput. Surv.* 54 (6): 1–42. https://doi.org/10.1145/3453474.

Guerrero, José, Luis Martí, Antonio Berlanga, Jesus Garcia, and José Molina. 2010. "Introducing a Robust and Efficient Stopping Criterion for MOEAs." July, 1–8. https://doi.org/10.1109/CEC.2010.5586265.

Halim, A. Hanif, I. Ismail, and Swagatam Das. 2021. "Performance Assessment of the Metaheuristic Optimization Algorithms: An Exhaustive Review." *Artif Intell Rev* 54 (3): 2323–409. https://doi.org/10.1007/s10462-020-09906-6.

Huband, S., P. Hingston, L. Barone, and L. While. 2006. "A Review of Multiobjective Test Problems and a Scalable Test Problem Toolkit." *IEEE Transactions on Evolutionary Computation* 10 (5): 477–506. https://doi.org/10.1109/TEVC.2005.861417.

Konak, Abdullah, David W. Coit, and Alice E. Smith. 2006. "Multi-Objective Optimization Using Genetic Algorithms: A Tutorial." *Reliability Engineering & System Safety* 91 (9): 992–1007. https://doi.org/10.1016/j.ress.2005.11.018.

Li, Miqing, and Xin Yao. 2020. "Quality Evaluation of Solution Sets in Multiobjective Optimisation: A Survey." *ACM Comput. Surv.* 52 (2): 1–38. https://doi.org/10.1145/3300148.

Liefooghe, Arnaud, and Bilel Derbel. 2016. "A Correlation Analysis of Set Quality Indicator Values in

Multiobjective Optimization." *Proceedings of the Genetic and Evolutionary Computation Conference 2016* (New York, NY, USA), GECCO '16, July, 581–88. https://doi.org/10.1145/2908812.2908906.

Liu, Yanfeng, Aimin Zhou, and Hu Zhang. 2018. "Termination Detection Strategies in Evolutionary Algorithms: A Survey." *Proceedings of the Genetic and Evolutionary Computation Conference* (New York, NY, USA), GECCO '18, July, 1063–70. https://doi.org/10.1145/3205455.3205466.

Madoumier, Martial, Gilles Trystram, Patrick Sébastian, and Antoine Collignan. 2019. "Towards a Holistic Approach for Multi-Objective Optimization of Food Processes: A Critical Review." *Trends in Food Science & Technology* 86 (April): 1–15. https://doi.org/10.1016/j.tifs.2019.02.002.

Martí, Luis, Jesús García, Antonio Berlanga, and José M. Molina. 2016. "A Stopping Criterion for Multi-Objective Optimization Evolutionary Algorithms." *Information Sciences* 367-368 (November): 700–718. https://doi.org/10.1016/j.ins.2016.07.025.

Perrignon, Manon, Thomas Croguennec, Romain Jeantet, and Mathieu Emily. 2024. "The Multi-Objective Data-Driven Approach: A Route to Drive Performance Optimization in the Food Industry." *Trends in Food Science & Technology* 152 (October): 104697. https://doi.org/10.1016/j.tifs.2024.104697.

Perrignon, Manon, Mathieu Emily, Mélanie Munch, Romain Jeantet, and Thomas Croguennec. 2025. "Machine Learning for Predicting Industrial Performance: Example of the Dry Matter Content of Emmental-Type Cheese." *International Dairy Journal* 162 (March): 106143. https://doi.org/10.1016/j.idairyj.2024.106143.

Roudenko, Olga, and Marc Schoenauer. 2004. "A Steady Performance Stopping Criterion for Pareto-Based Evolutionary Algorithms." April. https://inria.hal.science/hal-01909120.

Saxena, Dhish Kumar, Arnab Sinha, Joao A. Duro, and Qingfu Zhang. 2016. "Entropy-Based Termination Criterion for Multiobjective Evolutionary Algorithms." *IEEE Trans. Evol. Computat.* 20 (4): 485–98. https://doi.org/10.1109/TEVC.2015.2480780.

Tsarmpopoulos, Dimitris G., Athanasia N. Papanikolaou, Sotiris Kotsiantis, Theodoula N. Grapsa, and George S. Androulakis. 2019. "Performance Evaluation and Comparison of Multi-Objective Optimization Algorithms." *2019 10th International Conference on Information, Intelligence, Systems and Applications (IISA)* (PATRAS, Greece), July, 1–6. https://doi.org/10.1109/IISA.2019.8900773.

Wagner, Tobias, Heike Trautmann, and Boris Naujoks. 2009. *OCD: Online Convergence Detection for Evolutionary Multi-Objective Algorithms Based on Statistical Testing.* Vol. 5467. https://doi.org/10.1007/978-3-642-01020-0_19.

Wari, Ezra, and Weihang Zhu. 2016. "A Survey on Metaheuristics for Optimization in Food Manufacturing Industry." *Applied Soft Computing* 46 (September): 328–43. https://doi.org/10.1016/j.asoc.2016.04.034.

Zitzler, Eckart, Marco Laumanns, and Lothar Thiele. 2001. *SPEA2: Improving the Strength Pareto Evolutionary Algorithm.* ETH Zurich, Computer Engineering; Networks Laboratory. https://doi.org/10.3929/ethz-a-004284029.

