# OpenReview forum: "Comparative analysis of stopping criteria for multi-objective evolutionary algorithms: from benchmark problems to industrial application"
_Computo — Accepted by Computo_

### Review · Reviewer_kYwB · 2026-02-06

**Summary Of Contributions:**

The authors perform a comparison of four different stopping criteria for multi-objective evolutionary algorithms. They then propose a new criterion (the Maximum Performance Front), which should better balance computational cost and quality of the final Pareto front. Experimental results are performed on several established benchmarks and one industrial case study.

**Audience:**

Yes

**Broader Impact Concerns:**

No concerns.

**Claims And Evidence:**

Yes

**Requested Changes:**

# Major comments
Your description of the MPF on page 5 is a bit weird: the previous metrics have been introduced using a more mathematical notation, and suddenly the reader is faced with an algorithm. It's not bad per se, but maybe you could add an equation in the same style as Eq. (1), (2), (3), showing the new metric in a way which is more easily comparable to the old ones.

ZDT1 is the first in a series of well-known benchmarks (ZDT2 to ZDT6). Same for WFG, there are WFG1-9. Is there a reason you considered ZDT1 and WFG2-4 and ignored all the rest? I mean, the reason you picked each of the problems is stated on page 8, but why not consider the others as well? Are the other problems very similar to the instances you selected?

Also for the WFG problems, why did you only pick 2 and 4 objectives, skipping a configuration with 3? It is well known that MOEAs become less and less effective when the number of objectives increases, so it could have been interesting to see what happens with a large number of objectives, like 10. Note that I am not asking you to perform extra experiments, I would like to known the motivation between this selection of benchmark configurations.

If the number of pages allows it, it would be nice to have plots of the theoretical Pareto fronts for all the benchmarks (in the 2-objective case).

For Figures 2 and similar, I would like a short comment (maybe in the caption?) where you explain whether a high/low value is better, for each subfigure, to make it easier for the reader. Stopping generation: lower is better, ok. Same for Criterion Time and Overall Time. For Hypervolume, higher is better, right? And for Spread? Is it the same?

Another general comment on the figures: What is the unit of measurement for the two "Time" plots? At first I thought it was seconds, but there are negative values, is it the logarithm of the time? You should specify it.


# Minor comments
- page 2, line 34-35, there is a missing reference for NSGA-II
- line 39, "For the latter..." -> In the latter case, the algorithm keeps on...
- line 40, "An effective stopping criterion would be finding the minimal number of generations beyond which the algorithm..."
- line 47, Your definition of "convergence metrics" is a bit confusing. Instead of saying "Convergence metrics, on the other hand, evaluate how close the current solutions are to the theoretical Pareto front or a reference point in the objective space.", I would say: "Convergence metrics, on the other hand, either evaluate how close the current solutions are to the theoretical Pareto front, when this is known in advance; or evaluate the extension of the Pareto front with respect to a reference point in objective space."
- page 3, line 56, "Evaluation metric evaluates the evolution of the solutions" -> "The evaluation metric measures the evolution of the solutions...depending on the current value of the evaluation metric."
- line 72, "by Martì et al. (2016)"
- line 106, "10e-2" is a weird notation, I would either put "1e-2" or "$10^{-2}$"; same for line 169
- line 127, "iteration stop" -> "iterations stop"
- line 174, "which allows to compare" -> "which makes it possible to compare"
- line 179, missing whitespace after "."
- line 242, the first \hat{f}(x) should probably be \hat{f}_k(x), right?
- line 315, "with two objective" -> "with two objectives"
- a minor issue related to notation; in the caption of the figures, you used Y as the number of objective functions, while before you used k (see lines 242-244)
- line 336, "genaration" -> "generation"
- Figure 6, first "Hypervolume" subplot, I would like to see a zoom in, as clearly the wide variance in OCD_HV's performance hides the differences between the other techniques
- line 457, "convexe" -> convex

**Strengths And Weaknesses:**

Overall, the English of the paper is decent, with some sentences having a weird structure, but still understandable: see "minor issues" below for some examples of sentences which could be improved; after a while I stopped tracking them, there were too many instances. I would recommend passing the text through a LLM to smoothen it, or proofread it more deeply. Another general issue with formatting is that sometimes the references in the text are correctly reported between brackets, and sometimes they are not, when they obviously should be (see lines 72 and 79 for examples).

The topic and the main concepts needed for context are clearly introduced, and the experimental evaluation seems reasonable. The tendencies you identified with respect to the different termination criteria as the number of objectives grow are interesting, and I would have liked to see more (for example, what happens with 10 objectives, a number usually identified to be on the brink of the "many-objective" domain).

The main issue I see with the manuscript, is that I am not completely convinced that the experimental evidence supports the claim that MPF is actually better than the competition. This is due to: (i) some issues with figures (I don't understand the unit of measure for the 'y' axis of the plots for Time in Figures 2, 4, 6, 8; see Requested Changes); (ii) I think that the computational time for a criterion is a relatively minor issue in many industrial applications, where (from my own experience) evaluating a single solution can take up to minutes; (iii) the experiments on benchmarks only show results for 2 and 4 objectives, so I don't know how general the conclusions can be. I normally don't like it when reviewers ask for more experiments, but in this case I would be really interested to see if the trends you identified hold for 10 objectives, for example.

However, if the authors can provide some rebuttal for the points above, I am open to changing my mind and approve the paper for publication without the need of further experiments.

---

> ### Author Response · Authors · 2026-06-23
> **Response to reviewer comments (Part 1)– Manuscript Paper25**
>
> Thank you for giving me the opportunity to submit a revised version of my manuscript titled “Comparative analysis of stopping criteria for multi-objective evolutionary algorithms: from benchmark problems to industrial application” to Computo. We are grateful to the reviewer for the helpful comments they made on the original version of the manuscript. The present document provides a point-by-point response to the reviewer's comments and emphasizes the major modification of the revised manuscript. The revised manuscript is available on GitHub.
>
> **Comments on the overall structure:**
>
> Thank you for this comment. We have carefully reviewed the manuscript to ensure that all citations follow the required format. We also have revised the text throughout the manuscript to improve the clarity and quality of the English.
>
> **Comment (i):**
>
> Following the reviewer's suggestion, we have revised the figure captions to provide a clearer and more informative description of the figures presented.
>
> **Comment (ii):**
>
> We thank the reviewer for sharing this reflection. We acknowledge that in some industrial applications, the computational cost of the criterion is relatively minor. However, our work is motivated by real-time optimization contexts, where a solution must be selected and communicated rapidly to the operator in order to adjust the process accordingly. In this setting, the computational time required for the stopping criterion becomes relevant, since any delay in the decision-making process can directly impact operations. Therefore, we believe that favouring a criterion with low computational costs remains a practical consideration in such applications. To justify this choice and improve the reader's understanding of why criterion computation time is used as an indicator, we have added an explanation at line 204 of the revised manuscript.
>
> **Comment (iii):**
>
> We thank the reviewer for this comment and agree that extending the experimental evaluation to a higher number of objectives strengthens the generality of the conclusions. Following this suggestion, we conducted additional experiments for 6, 8 and 10 objectives. However, the computational cost increased for three criteria (OCD HV, LSSC and MPF) when 10 objectives were considered. This is due to the hypervolume computation, the calculation of which is known to scale exponentially with the number of objectives. This limitation is well documented in the literature and represents a constraint of hypervolume-based stopping criteria. The computational time required for one of the benchmark problems (WFG4) prevented the experiments from being completed within a reasonable computational time (for example, for an optimisation run, the computation time exceeded 40 hours to evaluate the five criteria using our PC’s configurations).
>
> Considering these observations, we chose to include the results for 8 objectives in the revised manuscript, as this setting provides a meaningful illustration of the criteria behavior as the number of objectives increases. Additionally, a dedicated paragraph has been added to the “Sensitivity of criteria to configuration parameters” section to explicitly address the limitations of the evaluated criteria in many-objective settings (> 10 objectives), providing the reader with a clearer picture of the boundaries of applicability of the proposed approach.

---

> > ### Author Response · Authors · 2026-06-23
> > **Response to reviewer comments (Part 2) – Manuscript Paper25**
> >
> > **Major comments**
> >
> > **Comment 1:**
> >
> > We thank the reviewer for this suggestion. We agree that the introduction of the MPF criterion lacked consistency with the notation used for the previous metrics. We have restructured and revised the relevant sections to improve clarity and coherence. In more details, we have added a dedicated section defining the evaluation metrics, followed by a novel section dedicated to the formal introduction of existing stopping criteria and our proposed MPF criterion. The modifications can be seen at line 97 of the revised manuscript.
> >
> > **Comment 2:**
> >
> > We thank the reviewer for this relevant question. Our benchmark selection was driven by the need to evaluate the stopping criteria across problems with a varying number of objectives and to cover a representative diversity of Pareto front and problem characteristics. The WFG benchmark problems was chosen for this purpose, as it is defined for any number of objectives, allowing us to compare the performance of criteria in settings with two, four and eight objectives (for the revised manuscript).
> > The ZDT benchmark problems, by contrast, is restricted to 2 objective problems. Moreover, the ZDT problems cover front type that are largely represented within the WFG suite: ZDT1 features a convex front, ZDT2 a concave front similar to WFG4 and ZDT3 a disconnected front, a property that is also present in WFG2. Including the full ZDT benchmark problems would therefore have introduced redundancy without adding new front diversity. ZDT1 was retained as a well-established 2-objective reference to complement the WFG results.
> > However, we agree that testing on a broader set of benchmarks would strengthen the robustness of our conclusions. This is an interesting direction for future work and a clarification has been added to the manuscript at line 591. We have also added an explanation in the “Multi-objective problem benchmarks” section.
> >
> > **Comment 3:**
> >
> > We thank the reviewer for this question. As discussed in our previous response, we agree on the importance of assessing criterion behavior with a large number of objectives and we have accordingly added experiments on WFG2, WFG3 and WFG4 with 8 objectives to the revised manuscript. With the inclusion of the eight-objective case, the manuscript now covers three configurations (2, 4 and 8 objectives) which we believe captures the key transitions in criterion behavior as the number of objectives increases.
> >
> > **Comment 4:**
> >
> > Following the reviewer's suggestion, we have added plots in appendix illustrating the theoretical Pareto fronts for all two-objective problems considered in the study.
> >
> > **Comment 5:**
> >
> > As suggested by the reviewer, we have improved the captions to make the presented metrics easier to understand.
> >
> > **Comment 6:**
> >
> > We thank the reviewer for quoting this inconsistency in the figures. Time was indeed displayed on a log scale and we have modified the figure captions accordingly.
> >
> > **Minor comments:**
> >
> > We thank the reviewer for their careful reading of the manuscript. All minor corrections have been addressed and incorporated into the revised version of the manuscript.

---

### Review · Reviewer_iZQg · 2026-05-11

**Summary Of Contributions:**

At this stage, this review is purely technical: following the withdrawal of one of the reviewers who had originally agreed to submit a report, we are moving on to the rebuttal phase with just one report. Two reviewers will carry out their work after the rebuttal

**Audience:**

Yes

**Broader Impact Concerns:**

see above

**Claims And Evidence:**

Yes

**Requested Changes:**

see above

**Strengths And Weaknesses:**

see above

---

### Comment · Editors_In_Chief · 2025-10-09
**Set up Continuous Intergration Appropriatly**

Dear authors,

Many thank you for submitting your work to Computo.

Before assigning an associated editor and send your manuscript for review, we need you to set up continuous integration so that your article manuscript is accessible via gh-pages.

Please refer to the comprehensive guide available on the following page to implement the CI process to ensure the reproducibility of your analyses.

https://computo-journal.org/site/guidelines-authors.html

Also feel free to ask for help by contacting us at contact@computo-journal.org

Regards,

---

### Comment · Action_Editor_NkGJ · 2026-05-11
**Rebuttal: discussion and revision period**

Dear authors,

We have received one report for your submission to Computo. Unfortunately, despite numerous reminders and a commitment to write a report, the second reviewer has failed to deliver.

We therefore propose moving on to the rebuttal period, as the first reviewer has provided relevant and constructive feedback that should help improve your submission. In a second stage, two reviewers will be asked once again to verify the changes you have made.

By default, a period of 6 weeks is allowed for discussion with the referees before they issue a final opinion. During this period, you can make any changes to your submission that you feel are necessary and that you are able to make. Hence, you can now respond to the first reviewer and make any changes to your submission that you deem necessary.

At the end of this period, the first reviewer - plus the second that has been secured for the second phase - will have about two weeks to make their decision, ranging from final acceptance to more substantial requests for modification. If you need an extra week or two, please let me know as soon as possible.

Best regards

---

### Note · Reviewer_kYwB · 2026-07-03

**Comment:**

The authors replied in a satisfying way to all my inquiries, and I think that the manuscript is now ready for publication in its current state.

**Audience:**

Yes

**Claims And Evidence:**

Yes

**Decision Recommendation:**

Accept

---

### Note · Reviewer_iZQg · 2026-07-08

**Comment:**

The authors have adequately addressed the scientific questions raised. The issue of reproducibility remains, and this will be discussed by the associate editor prior to final acceptance.

**Audience:**

Yes

**Claims And Evidence:**

Yes

**Decision Recommendation:**

Accept

---

### Decision · Action_Editor_NkGJ · 2026-07-08

**Recommendation:** Accept with minor revision

**Comment:**

# Reproducibility Audit

The article is scientifically validated, but the repository does not yet meet the journal's reproducibility standards. We'd like to ask the authors to address the points below before the repository is transferred to `computorg`.

## Reproducibility issues

### 1. The manuscript does not recompute anything

The `.qmd` file contains chunks that essentially call `readRDS()` on pre-computed `.rds` files (`tables/*.rds`) and display static images (`figures/*.png`). The actual computation (NSGA-II algorithm, stopping criteria, industrial case) lives in a separate script, `script/script_optimization.R` (2027 lines), which is never invoked, either by the `.qmd` or by CI.

As a result, the current CI only verifies that the manuscript compiles from already-frozen results, not that the scientific pipeline itself is reproducible — which is central to Computo's editorial policy.

Is there a specific reason the code isn't run as part of the build (e.g., runtime too long, or the industrial use case)? If so, we'd suggest integrating at least part of the computation into the `.qmd` to demonstrate reproducibility — for instance on a toy example with characteristics similar to the full dataset. For the remaining, more demanding computations, it would help to store the raw results and include the code that regenerates at least the plots and tables from those results, so the link between code and outputs is traceable even where full re-execution isn't practical.

### 2. The lines that generate the cached files are commented out

In `script/script_optimization.R`, the `write_rds(...)` calls that would produce the `.rds`/tables used by the article are currently commented out (e.g. lines 614, 664, 684, 767, 789, 812, 835, 1063, 1086). As it stands, even running the script manually wouldn't regenerate anything without first editing the code.

Could you uncomment or restructure these calls so the script actually produces the artifacts it's meant to produce?

### 3. Hardcoded Windows absolute path

`script/script_optimization.R:1910`:
```r
read_rds("C:/Users/manon/Documents/GitHub/Computo_MPerrignon/data/industrial_case/results_industrial_case.rds")
```
This path only works on the lead author's machine and would break execution anywhere else (including in CI). Please replace it with a project-relative path.

### 4. Git repository carrying R session files and large data (149 MB `.git`)

- `script/.RData`: 81 MB, tracked.
- `.RDataTmp` (root): 28 MB, tracked.
- `.Rhistory` (root and `script/`): tracked — session files with no scientific value.
- `data/industrial_case/modY1.rds` … `modY4.rds`, `results_HV.rds`: 27–48 MB each, tracked directly in git.

It looks like the repository's `.gitignore` is missing the standard block used by other already-published articles (`.Rhistory`, `.RData`, `.Ruserdata`, `renv/`, `_cache`, `_freeze`, `*_files`, etc.), which would explain how these ended up versioned.

To address this, we'd suggest:

- Adopting the journal's standard `.gitignore`.
- Removing `.RData`, `.RDataTmp`, `.Rhistory` from git tracking (`git rm --cached`) and cleaning up the history before the repository is transferred to `computorg` — the repo will likely be recreated/transferred at that point anyway, which is a natural opportunity to start from a clean history.
- Moving the large industrial-case `.rds` files to an external data repository (Zenodo/OSF, with a DOI), per the journal's policy that large data files can be stored externally, and keeping only lightweight example data (or a download script) in the repo itself.

### 5. No data/code availability statement in the manuscript

We couldn't find a "Data availability" or "Reproducibility" section in the `.qmd`. The industrial-case data (dairy process models from STLO/INRAE) seem to have a particular status (industrial confidentiality?) that would be worth making explicit in the article, rather than shipping `.rds` files whose provenance isn't documented.

Could you add a short section clarifying the status of the data (public/simulated vs. confidential industrial) and how each part of the study can be reproduced?

---

## Minor/technical suggestions

### 6. `_quarto.yml` metadata not updated from the template

- `repo: "template-computo-r"` (line 48) looks like a template leftover — it should point to the actual repository.
- The project's render list references `README.qmd`, which doesn't exist (only `README.md` is present). We'd also suggest updating to the latest version of the Computo template and extension.

### 7. Unused `environment.yml`

This conda environment installs Python/Jupyter/Plotly/Kaleido/Pandas/Numpy, but nothing in the `.qmd` or the R script uses Python. Likely a template leftover — removing it would avoid misleading signals about what's actually needed to reproduce the analysis.

---

## Summary

1. Make the `.qmd` executable end-to-end where feasible (live computation or via Quarto cache), and for the heavier computations, at least include the code that regenerates plots/tables from stored raw results.
2. Uncomment the `write_rds(...)` calls and remove the hardcoded Windows path.
3. Clean up the git repository: standard `.gitignore`, removal of `.RData`/`.RDataTmp`/`.Rhistory`, externalization of large `.rds` files to Zenodo/OSF with a DOI.
4. Add a "Data availability" section to the manuscript.
5. Clean up template leftovers (`repo:` in `_quarto.yml`, the `README.qmd` reference, the unused `environment.yml`).

Items 1–4 should be addressed before publication; item 5 is minor but should also be fixed before the repository is transferred to `computorg`.

**Audience:**

Fall in the journal's perimeter.

**Claims And Evidence:**

yes, see reviewers' comments.

---

> ### Decision · Editors_In_Chief · 2026-07-08
>
> I approve the AE's decision.